# ELICITING NUMERICAL PREDICTIVE DISTRIBUTIONS OF LLMS WITHOUT AUTOREGRESSION

**Julianna Piskorz**[*], **Katarzyna Kobalczyk**,[*] **Mihaela van der Schaar**
Department of Applied Mathematics and Theoretical Physics
University of Cambridge
`{jp2048,knk25}@cam.ac.uk`

## ABSTRACT

Large Language Models (LLMs) have recently been successfully applied to regression tasks—such as time series forecasting and tabular prediction—by leveraging their in-context learning abilities. However, their autoregressive decoding process may be ill-suited to continuous-valued outputs, where obtaining predictive distributions over numerical targets requires repeated sampling, leading to high computational cost and inference time. In this work, we investigate whether distributional properties of LLM predictions can be recovered *without* explicit autoregressive generation. To this end, we study a set of regression probes trained to predict statistical functionals (e.g., mean, median, quantiles) of the LLM's numerical output distribution directly from its internal representations. Our results suggest that LLM embeddings carry informative signals about summary statistics of their predictive distributions, including the numerical uncertainty. This investigation opens up new questions about how LLMs internally encode uncertainty in numerical tasks, and about the feasibility of lightweight alternatives to sampling-based approaches for uncertainty-aware numerical predictions. Code to reproduce our experiments can be found at `https://github.com/kasia-kobalczyk/guess_llm.git`.

## 1 INTRODUCTION

With the increasing capabilities of LLMs, a growing body of work has explored their use for structured data prediction—most notably for tabular data regression (e.g. Requeima et al., 2024; Hegselmann et al., 2023; Shysheya et al., 2025; Vacareanu et al., 2024) and time series forecasting (e.g. Gruver et al., 2024; Xue & Salim, 2023). These studies demonstrate that LLMs can act as competitive regressors, even without task-specific fine-tuning. This advantage is especially pronounced in low-data regimes, where LLMs can leverage their pre-training, prior knowledge, and capacity to condition on auxiliary textual context, matching or outperforming specialised models. We provide a more detailed overview of related works in Appendix A.

However, issuing numerical predictions with LLMs requires sequential autoregressive generation: real-valued numbers typically span multiple tokens requiring multiple forward passes of the input through the LLM to generate a single prediction. This makes inference costly and time-consuming, especially when many samples are needed—for example, to quantify the uncertainty (Gruver et al., 2024; Requeima et al., 2024) or improve prediction accuracy (Requeima et al., 2024).

This raises a natural question: is there a way of eliciting the LLM's predictive distribution, and its underlying uncertainty, without performing costly autoregressive number generation and repeated sampling? This question is non-trivial: generating a real number requires determining its order of magnitude, including decisions about decimal placement and number termination–choices that are typically made only after several tokens have already been generated. By exploring whether we can gain insight into such decisions without performing autoregressive token generation, our work contributes to exploring the broader question of how LLMs 'plan' their outputs before generation begins (Lindsey et al., 2025).

---

[*]Authors contributed equally

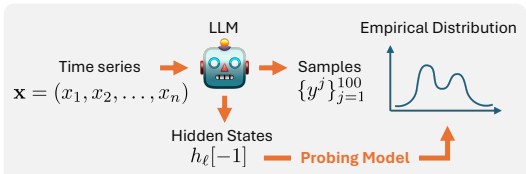 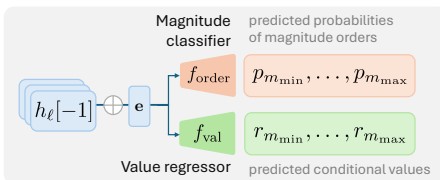

(a) The goal: can we recover the LLM's numerical predictive distribution from its internal representation of the input?

(b) Schematic diagram of the magnitude-factorised probing model used in section 2.

Figure 1: Illustration of this paper's goals and methodology.

In this work, focusing on the problem of time series forecasting specifically, we explore to what extent the LLM's internal representation of the input sequence (requiring just a single pass through the LLM) can be used to reconstruct its predictive distribution of the next real number. Concretely, we explore the following three questions:

**Do LLMs encode the next number they intend to generate? (Section 2)** We begin by examining whether LLM's internal representations of the input series encode sufficient information to recover point predictions–specifically, the greedy output, mean, and median of the predictive distribution. To test this, we develop **a magnitude-factorised regression probe** that separates prediction into two components: a coarse magnitude classification and a scale-invariant value regression, such that our model can effectively learn to predict numbers of varying orders of magnitude. Trained on LLM embeddings from synthetic time series data, *our probe accurately predicts numerical targets across data with varying orders of magnitude*.

**Can we elicit the uncertainty of the LLM's predictive distribution? (Section 3)** We then ask whether uncertainty information is also captured in LLM's hidden states. Using a similar magnitude-factorised quantile regression model, we train probes to predict the quantiles of the LLM's output distribution, approximated via sampling. *The resulting models accurately recover the interquartile range and produce well-calibrated confidence intervals*.

**What is the practical value of our findings? (Sections 4 and 5)** The ability to recover numerical predictions directly from LLM embeddings holds the potential to bypass autoregressive sampling— offering savings in inference time and computational costs, which we demonstrate empirically. However, for such probes to be practically useful, they must generalise beyond the specific conditions under which they were trained. We therefore evaluate whether a single probing model can be deployed across varied settings without retraining. First, we test generalisation to unseen time series lengths. Second, we assess generalisability of our previous results to real-world data. We investigate whether probes trained on real-world data generalise across different sub-domains and whether probes trained on synthetic data generalise to real-world data. *We demonstrate that, while some drop in calibration occurs on out-of-distribution datasets, our probes demonstrate encouraging generalisation abilities, showing the universality of LLM's representations of numerical quantities*.

Our findings provide new insights into the numerical capabilities of LLMs: much of the "reasoning" underlying numerical predictions appears to be encoded in the model's internal representations of the input, prior to token-level decoding. This contests whether autoregressive sampling is necessary to extract real-valued outputs from LLMs, and opens the door to developing more efficient, single-pass approaches. By showing that both point estimates and uncertainty can be reliably extracted from hidden states, our work suggests a lightweight, general-purpose strategy for deploying LLMs in regression tasks–particularly in settings where computational efficiency and uncertainty estimation are essential. We hope these results motivate further study of how LLMs internally represent numerical quantities, and how this information can be surfaced for practical downstream use.

## 2 DO LLMS ENCODE THE NEXT NUMBER THEY INTEND TO GENERATE?

LLMs are trained for next-token-predictions. Thus, as a single number typically spans multiple tokens, obtaining a complete numerical prediction from the LLM requires repeated auto-regressive sampling. This can be computationally expensive in number-heavy tasks, particularly when one would like to obtain repeated samples for the purpose of uncertainty estimation. To mitigate this overhead, we ask: *to what extent is the full number the LLM intends to predict–beyond just its leading*

*digit–already encoded in the LLM's internal representation, prior to any token-by-token generation?* If such information can be reliably extracted, one could sidestep autoregressive generation altogether. However, this possibility is not trivial: critical aspects of number generation, such as the placement of the decimal point or number termination, which determine the order of magnitude of the number, often occur late in the decoding process, particularly for large magnitudes.

## 2.1 METHOD

**Objective.** Let $\mathbf{x} = [x_1, \ldots, x_n]$ be a sequence of numbers (e.g., an equally-spaced time series). Given $\mathbf{x}$, a language model induces a predictive distribution $p_{\text{LLM}}(\cdot \mid \mathbf{x})$ over the next value $x_{n+1}$. In this section, we investigate whether the internal representations of the LLM encode sufficient information to predict this distribution's key statistics. Specifically, we aim to train independent probing models to recover: (a) the LLM's greedy prediction, (b) the mean, and (c) the median of $p_{\text{LLM}}$. The empirically estimated *scalar* statistics of the LLM's predictive distribution are our targets for prediction based on the LLM's hidden representation of the input series $\mathbf{x}$.

**LLM Representation.** Following Gruver et al. (2024), we serialise an input series $\mathbf{x}$ to text as "$x_1, x_2, x_3, \ldots, x_n,$". We *do not* apply any scaling to the time series before serialising the inputs. This is important, as LLMs often contain contextual prior knowledge and scaling of the original time series may prohibit the LLM from using this prior knowledge effectively. From a pre-selected set of $N$ transformer layers denoted by $\mathcal{H}$, we extract the final token's hidden state from each layer, obtaining a set of embeddings $\{\mathbf{h}_\ell[-1] \in \mathbb{R}^{d_{\text{model}}} : \ell \in \mathcal{H}\}$. We concatenate these embedding vectors to form a single input for the probe:

$$\mathbf{e} := \text{concat} \left(\mathbf{h}_\ell[-1]\right)_{\ell \in \mathcal{H}} \in \mathbb{R}^{d_{\text{input}}}, \tag{1}$$

where $d_{\text{input}} = d_{\text{model}} \times |\mathcal{H}|$. The choice of the hidden layers $\mathcal{H}$ is a hyperparameter of our model. Throughout the main body of this paper we use the Llama-2-7B model, for which $d_{\text{model}} = 4096$, and we set $\mathcal{H}$ as the last 8 layers. This choice of model is motivated by the fact that the tokeniser of Llama-2-7B encodes each digit as a separate token, thus allowing to ensure that numbers with larger orders of magnitude consist of more tokens (making it more difficult to obtain LLM's predictions without autoregressive decoding). Results with other LLMs can be found in Appendix C, alongside an ablation study on the choice of layers.

**Datasets.** We use synthetically generated datasets to evaluate probing performance. Each sequence $\mathbf{x}$ is sampled from a set of functions exhibiting varied dynamics, including sinusoidal patterns, Gaussians, beat functions, and random noise (see Appendix B.3 for details). The generated time series also vary in the length, $n$, and the level of noise, to ensure diversity of the input embeddings and target distributions. We generate variants of the dataset by scaling the value range from $[-1, 1]$ to $[-10, 10]$, $[-1000, 1000]$, and $[-10000, 10000]$. We also combine the datasets of different scales to obtain a larger dataset of approximately 66k unique sequences, balanced across the different orders of magnitude. In this section, our training datasets take the following structure: $\left\{ \left(\mathbf{x}_i, \mathbf{e}_i, y_i^{\text{greedy}}, y_i^{\text{mean}}, y_i^{\text{median}}\right) \right\}_{i=1}^N$, where $N$ is the total number of examples in a dataset, $y_i^{\text{greedy}}$ the LLM's greedy prediction given $\mathbf{x}_i$, and $y_i^{\text{mean}}, y_i^{\text{median}}$ the empirical mean and median, respectively, estimated based on $N_{sa}$ samples from the LLM's predictive distribution, $\{y_i^j\}_{j=1}^{N_{sa}} \sim p_{\text{LLM}}(\cdot | \mathbf{x}_i)$. In our experiments we set $N_{sa} = 100$.

**Probing Model.** To the best of our knowledge, existing approaches to neural network probing are limited to binary or categorical probes (see Appendix A). A significant challenge in training *regression* probes for LLM numerical predictions is the wide spread of target magnitudes. Standard regression losses such as the MSE, or transformation techniques like log-scaling, fail to provide stable gradients, prioritising optimisation of the largest values only. To address this, we introduce a *magnitude-factorised regression model* that consists of two components (both initialised as MLPs, see Appendix B.4 for hyperparameter details):

- $f_{\text{order}} : \mathbb{R}^{d_{\text{input}}} \to \mathbb{R}^M$: a classifier predicting the order of magnitude of the target number.
- $f_{\text{val}} : \mathbb{R}^{d_{\text{input}}+1} \to \mathbb{R}^M$: a regressor predicting the target value scaled by the predicted order.

The magnitude classification component predicts the order of magnitude of a target value $y^*$, where $* \in \{\text{greedy}, \text{mean}, \text{median}\}$. Formally, we define the order of magnitude of a scalar $y$ as: $m(y) =$

$\lfloor \log_{10}(|y|) \rfloor$. For a given input $\mathbf{x}$ and its corresponding representation $\mathbf{e}$, $f_{\text{order}}(\mathbf{e})$ outputs a vector of logits of a set of pre-defined magnitude classes $m_k \in \{m_{\min}, \ldots, m_{\max}\}$. In our experiments, we let $m_{\min}$ and $m_{\max}$ be the minimum and maximum orders of magnitude in the training data. The predicted vector of magnitude class probabilities (after softmax) is denoted as $\text{softmax}(f_{\text{val}}(\mathbf{e})) = \mathbf{p}(\mathbf{x}) = [p_{m_{\min}}, \ldots, p_{m_{\max}}]$.

The regression component predicts a scaled version of the target value, conditioned on all possible magnitude classes (to allow the model to adjust its prediction based on the predicted magnitude). For each magnitude class $m_k \in \{m_{\min}, \ldots, m_{\max}\}$, the corresponding scale is equal $s_k = 10^{m_k}$. The regression head takes as input the concatenation of the feature representation $\mathbf{e}$ and the scale factor $s_k$ outputting $r_k = f_{\text{val}}([\mathbf{e}; s_k])$ for each $k$. This produces regression outputs $\mathbf{r} = [r_{m_{\min}}, \ldots, r_{m_{\max}}]$ for all magnitude classes. The conditional prediction for each magnitude order $m_k$ is then computed as: $\hat{y}_k = r_k \cdot 10^{m_k}$.

**Loss Function and Training.** Our model supports a two-phase training procedure:

- **Phase 1:** Train only the classification head while freezing the regression head, using the classification loss—standard cross-entropy loss for magnitude prediction:

$$\mathcal{L}_{\text{order}} = \frac{1}{N_b} \sum_{i=1}^{N_b} \text{CrossEntropyLoss}(\mathbf{p}(\mathbf{x}_i), m(y_i^*)), \tag{2}$$

  where $N_b$ is the batch size.

- **Phase 2:** Train only the regression head while freezing the classification head, using the regression loss, i.e. the mean squared error between the predicted scaled value and the scaled target value:

$$\mathcal{L}_{\text{val}} = \frac{1}{N_b} \sum_{i=1}^{N_b} \left( r_{\hat{m}_i} - \frac{y_i^*}{10^{m(y_i^*)}} \right)^2, \quad \text{where} \quad \hat{m}_i = \arg\max_k p_k(\mathbf{x}_i). \tag{3}$$

Empirically, we find that the 2-stage training procedure performs better than joint training of the order and value heads. To further provide justification for our approach, in Appendix C.1 we compare the performance of our magnitude-factorised probe against a vanilla MLP probe, showing significant performance gains.

**Expected Prediction.** During evaluation of our model, we compute the *expected prediction* by marginalising over the top-$K$ magnitude classes: $\mathbb{E}_K[\hat{y}] = \sum_{k \in \text{top-}K} p_k \hat{y}_k$, where top-$K$ refers to the $K$ magnitude classes with highest predicted probabilities (we set $K = 3$).

## 2.2 RESULTS

**Order of magnitude.** We first investigate to what extent our probing model can correctly recover the order of magnitude of the number the LLM intends to generate. We train three separate models, one for each of the mean, median and greedy targets. In this experiment, we use the combined dataset consisting of time series with varying scales. The bar chart on the right hand side of Figure 2 visualises that the classifier part of our magnitude-factorised model achieves above 90% accuracy in predicting the exponent of the target values. Further, as visualised with the scatter plots on Figure 2, we find strong correlation between our probes' final predictions and the target statistics.

**Precision in generated digits.** To further assess whether the LLM's internal representations encode fine-grained information beyond the order of magnitude, we focus on the dataset with time series values in the interval $[-1, 1]$. We report the mean squared error (MSE) of the predictions obtained with our probing model in Table 1. For context, we compare the performance of our probe against three simple base-lines, using as the prediction: a) the average value of the entire training dataset ($\bar{\mathbf{x}}$), b) the average of the series $\mathbf{x}_i$ ($\bar{\mathbf{x}}_i$), or c) the last value of the series $\mathbf{x}_i$ ($x_{i,n}$). We also provide scatter plots for this dataset in Appendix C. The results demonstrate that using the last token embeddings of the LLM as the input, our probe can accurately recover the LLM's predictions, at an accuracy level which

Table 1: MSE obtained when predicting the statistics of the LLM's predictive distribution.

| $y_i^*$, target | $\hat{y}_i^*$ (ours) | $\bar{\mathbf{x}}$ | $\bar{\mathbf{x}}_i$ | $x_{i,n}$ |
|---|---|---|---|---|
| $* = $ mean | 0.006 | 0.256 | 0.035 | 0.085 |
| $* = $ median | 0.006 | 0.260 | 0.041 | 0.087 |
| $* = $ greedy | 0.015 | 0.273 | 0.065 | 0.109 |

Figure 2: *Predicted vs. true values of mean, median and greedy prediction, presented on $\log_{10}$ scale. The probing model accurately recovers the number that the LLM intends to predict, indicating that the internal representations encode the order of magnitude of prediction.*

significantly surpasses simple baselines naively constructed from the input. Interestingly, among the three targets considered (mean, median, greedy), the model performs worst when predicting the greedy output. We hypothesise that this is because the greedy prediction is not an explicit function of the model's predictive distribution, but rather a by-product of the autoregressive decoding process, making it harder to recover precisely from internal states.

> 💡 These results show that **the internal representations of a pre-trained LLM encode detailed information about its intended numerical output**—even before any tokens are generated. Our probing model accurately recovers not only the order of magnitude, but also fine-grained point estimates of the mean, median, and the greedy output. This demonstrates that much of the numerical reasoning performed by the LLM is already present in its hidden states, and may not require the autoregressive decoding process.

## 3   CAN WE ELICIT THE UNCERTAINTY OF THE LLM'S PREDICTIVE DISTRIBUTION?

In the previous section, we demonstrated that point estimates—such as the greedy prediction, mean, and median—of an LLM's predictive distribution $p_{\text{LLM}}(\cdot \mid \mathbf{x})$ can be recovered from its internal representations without the need for performing autoregressive sampling. Encouraged by these findings, we now investigate whether we can go beyond point estimates to recover the *uncertainty* of $p_{\text{LLM}}$ by approximating its distributional shape. Specifically, we attempt to recover multiple quantiles of $p_{\text{LLM}}$, enabling a coarse-grained reconstruction of its distribution function and providing an easy way of estimating the confidence intervals for the LLM's predictions.

### 3.1   METHOD

**Quantile Regression.** Since the distribution $p_{\text{LLM}}$ may be multi-modal and non-Gaussian, we rule out parametric approximations. Instead, we adopt *quantile regression*, which enables direct estimation of distributional shape without strong assumptions about its form. Let $\mathcal{Q} = [\tau^1, \dots, \tau^S]$ be a list of target quantile levels. For each $\tau^s \in [0, 1]$, we denote the predicted quantile value as $\hat{q}^s$. We train the quantile predictor using the *pinball loss* (Koenker & Hallock, 2001), computed with respect to LLM samples $y_i^j \sim p_{\text{LLM}}(\cdot \mid \mathbf{x}_i)$. For a single quantile level $\tau$, predicted quantile value $\hat{q}$ and a single LLM sample $y_i^j$, this loss function is defined as:

$$\text{PinballLoss}(\tau, \hat{q}, y_i^j) := \max\left(\tau(y_i^j - \hat{q}), (1 - \tau)(\hat{q} - y_i^j)\right). \tag{4}$$

**Probing Model.** As in section 2, we use a magnitude-factorised model to address the challenge of scale variance in numerical outputs. The quantile model takes an equivalent form to the one in section 2, except we introduce $S$ classification and regression heads, one for each quantile level, denoted by $f_{\text{order}}^s$ and $f_{\text{val}}^s$, respectively. As previously, each classification head outputs a vector of magnitude class probabilities $\mathbf{p}^s = [p_{\min}^s, \dots, p_{\max}^s]$. The regression heads output vectors of conditional scaled values $\mathbf{r}^s = [r_{\min}^s, \dots, r_{\max}^s]$. For an order $m_k$ the predicted conditional quantile value is computed as $\hat{q}^s = r_k^s \cdot 10^{m_k}$.

**Datasets.** We use the same datasets as in section 2, but with target values being the raw LLM samples $\{y_i^j\}_{j=1}^{N_{sa}}$, instead of the aggregate statistics.

**Training.** Unlike in the previous section, we use a joint training approach (2-phase training did not yield any significant improvements in performance). We construct the total loss as the sum of the cross-entropy losses for magnitude prediction and pinball losses for quantile regression:

$$\mathcal{L} = \sum_{s=1}^{S} \left( \mathcal{L}_{\text{order}}^{s} + \beta \cdot \mathcal{L}_{\text{val}}^{s} \right), \tag{5}$$

$$\mathcal{L}_{\text{order}}^{s} = \frac{1}{N_b} \sum_{i=1}^{N_b} \text{CrossEntropyLoss} \left( \mathbf{p}^{s}(\mathbf{x}_i), m(\tilde{q}_i^{s}) \right), \tag{6}$$

$$\mathcal{L}_{\text{val}}^{s} = \frac{1}{N_b N_{sa}} \sum_{i=1}^{N_b} \sum_{j=1}^{N_{sa}} \text{PinballLoss} \left( \tau^{s}, r_i^{s}, \frac{y_i^{j}}{10^{m(\tilde{q}_i^{s})}} \right). \tag{7}$$

In the above, $\tilde{q}_i^{s}$ denotes the empirical quantile value derived from the LLM samples $\{y_i^{j}\}_{j=1}^{N_{sa}}$. In our experiments, we use a set of $S = 7$ quantile levels: $\mathcal{Q} = [0.025, 0.05, 0.25, 0.5, 0.75, 0.95, 0.975]$. This choice of allows us to easily estimate: the median, the interquartile range (IQR), as well as the 90% and 95% confidence intervals.

## 3.2 RESULTS

**IQR Prediction.** To investigate whether the LLM's internal representations encode information about the spread of its predictive distribution, we estimate the interquartile range (IQR) using the predicted 25th and 75th percentiles. As the IQR is sensitive to scale, we normalise it by the predicted median, and similarly normalise the empirical IQR from LLM samples using the sample median. If the probe captures uncertainty faithfully, we should observe a monotonic relationship between the predicted and empirical (normalised) IQRs. Scatter plots in Figure 3 show a strong correlation between predicted and sample-based IQRs, with points aligning well on $y = x$. This demonstrates that our probing model is able to infer distributional spread from the LLM's hidden states.

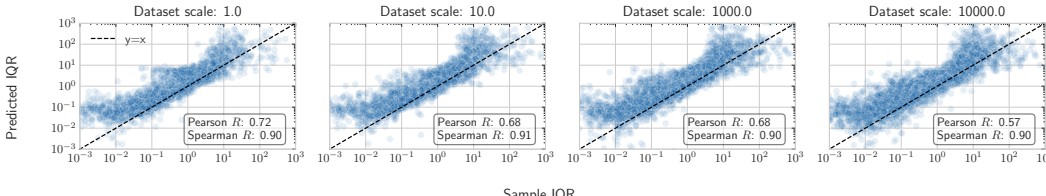

Figure 3: Predicted vs. sample-based IQR (both median-normalised). The model accurately tracks the variability of the LLM's output distribution.

**Confidence Interval Coverage.** Next, we evaluate whether the predicted quantiles yield calibrated confidence intervals. Given a desired confidence level $\alpha$ and its associated interval $\mathcal{C}(\alpha)$ predicted by the probe, we compute the empirical coverage by checking what fraction of LLM samples fall within the predicted interval. We expect that:

$$\alpha = \mathbb{E}_{y \sim p(\cdot|\mathbf{x})} \left[ \mathbb{1}\{y \in \mathcal{C}(\alpha)\} \right]$$

$$\approx \frac{1}{N_{sa}} \sum_{j=1}^{N_{sa}} \mathbb{1}\{y^{j} \in \mathcal{C}(\alpha)\}, \quad \text{where } y^{j} \sim p_{\text{LLM}}(\cdot|\mathbf{x}).$$

Table 2: Coverage of the predicted confidence intervals. Values denote empirical coverage (%) ± standard error.

| $\alpha$ | 50% | 90% | 95% |
|---|---|---|---|
| 1.0 | $52.0 \pm 0.4$ | $90.9 \pm 0.3$ | $95.5 \pm 0.2$ |
| 10.0 | $52.7 \pm 0.5$ | $91.3 \pm 0.3$ | $96.1 \pm 0.2$ |
| 1000.0 | $51.4 \pm 0.3$ | $90.7 \pm 0.3$ | $95.7 \pm 0.2$ |
| 10000.0 | $48.2 \pm 0.3$ | $90.5 \pm 0.2$ | $95.4 \pm 0.2$ |

Table 2 reports the empirical coverage for 50%, 90%, and 95% intervals across datasets of varying scale. In all cases, empirical coverage closely matches the target level, indicating that the quantile probe is well-calibrated.

> 💡 Our findings provide strong evidence that **the uncertainty of the LLM's predictive distribution is encoded in its internal activations and can be effectively elicited using a quantile regression probe**. The probe is capable of predicting meaningful spread measures, producing well-calibrated confidence intervals that match the empirical coverage. These results suggest that LLMs internalise rich distributional information during generation, which can be accessed and approximated efficiently via probing. This opens up new opportunities for downstream applications that rely on uncertainty quantification—such as safe decision-making, model-based control, and probabilistic reasoning—while avoiding the overhead of autoregressive sampling.

## 4    EFFICIENCY AND ACCURACY

In this section, we further investigate the practical applicability of probing models from the perspective of computational and inference time efficiency as well as accuracy with respect to the ground truth time series value. Throughout this section, we focus on the scalar probing models from section 2.

### 4.1    ERRORS WITH RESPECT TO THE GROUND-TRUTH

Table 3: MSE of predicted values vs. the ground truth $x_{i,n+1}$.

| predicted value | probe ($\hat{y}_i^*$) | LLM ($y_i^*$) |
|---|---|---|
| $* =$ greedy | 0.0652 | 0.0668 |
| $* =$ mean | 0.0562 | 0.0555 |
| $* =$ median | 0.0561 | 0.0553 |

| $\bar{\mathbf{x}}$ | $\bar{\mathbf{x}}_i$ | $x_{i,n}$ | GP |
|---|---|---|---|
| 0.3454 | 0.0878 | 0.1226 | 0.0717 |

In this experiment, we compare the error of our probes' predictions with respect to the ground-truth next value $x_{i,n+1}$ of the input time series $\mathbf{x}_i = (x_{i,1}, \ldots, x_{i,n})$. We present the comparison between the error achieved by using the probe trained to predict the mean, median and greedy statistics of the LLM distribution ($\epsilon_{\text{probe}}^* = \mathbb{E}\left[(\hat{y}_i^* - x_{i,n+1})^2\right]$) vs. by using these statistics directly ($\epsilon_{\text{LLM}}^* = \mathbb{E}\left[(y_i^* - x_{i,n+1})^2\right]$). For a better sense of scale, we additionally present the errors using the simple baselines introduced in section 2.2, as well as the error of the mean prediction of a GP model fit to the input time series $\mathbf{x}_i$. The MSE values of the predictions with respect to the ground-truth next value of the time series are presented in Table 3. Results were obtained for the probing models from section 2.2 and the dataset with scale 1.0. Our probe attains errors comparable to those obtained by sampling directly from the LLM. This places its performance in context: **the probe captures enough information from hidden states to match the LLM's own accuracy on the one-step-ahead prediction task.**

### 4.2    SAMPLE EFFICIENCY AND COMPUTATIONAL COSTS

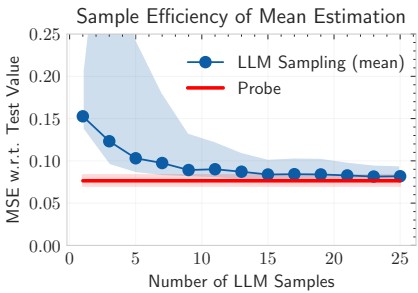

Figure 4: The probe achieves comparable error to using 20-25 LLM samples on the one step ahead prediction task.

We further compare the error of the probe in predicting $x_{i,n+1}$ vs. using the mean prediction of the LLM using $N$ empirical samples. Figure 4 illustrates this on a dataset with scale 1.0. The horizontal line shows the error attained by our probe, $\epsilon_{\text{probe}}^{\text{mean}} = \mathbb{E}\left[(\hat{y}_i^{\text{mean}} - x_{i,n+1})^2\right]$ and the blue points with error bars show the error attained when using $N$ LLM samples, $\epsilon_{\text{LLM}}^{\text{mean}}(N) = \mathbb{E}\left[(y_i^{\text{mean},N} - x_{i,n+1})^2\right]$, where $y_i^{\text{mean},N}$ is the mean of $N$ LLM samples: $\{y_i^j\}_{j=1}^N \sim p_{\text{LLM}}(\cdot|\mathbf{x}_i)$. Our probe outperforms empirical sampling for all $N$ up to **20-25 samples**, demonstrating that a probe of this kind can serve as a **computationally efficient surrogate for making numerical predictions**.

For a detailed discussion regarding the computational costs comparisons of LLM sampling and inference with our probe, see Appendix C.3.

> 💡 **Key Insights.** The probes trained to predict the LLM's output from its hidden state achieve high enough precision to be used as competitive predictors on the one-step-ahead time series prediction task. This demonstrates that they can be used as a lightweight way of obtaining LLM predictions, delivering **comparable performance at substantially lower inference cost**.

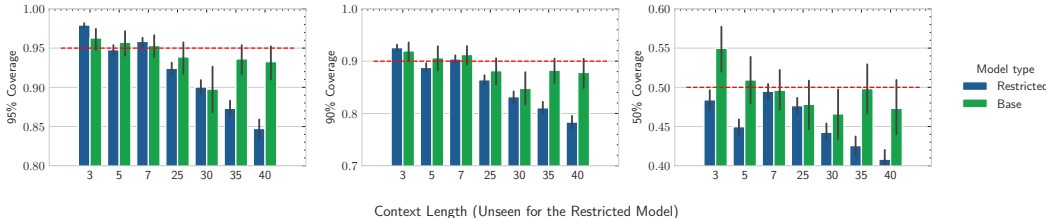

Figure 5: *Generalisation to unseen context lengths.* A probe trained on a restricted context length range (Restricted) exhibits greater deviation in empirical coverage outside its training range.

## 5 GENERALISATION

Finally, we investigate the generalisation capabilities of our approach along several axes including context length generalisation, applicability to real-world data, and cross-dataset generalisation. As the process of training a probe can be costly, such generalisation capabilities are important for real-world applications, if we would like to use a pre-trained probe on new datasets with different distributional properties. Throughout this section, we focus on the quantile probing models from section 3.

### 5.1 GENERALISATION TO UNSEEN CONTEXT LENGTHS

First, we ask whether a probe trained on a fixed range of input sequence lengths generalises to longer or shorter contexts. We train and compare against each other two models:

- **Base:** Trained on input sequences $\mathbf{x}$ with lengths in range $[3, 40]$.
- **Restricted:** Trained only on input sequences $\mathbf{x}$ with lengths in range $[10, 20]$.

At test time, we evaluate both models on contexts shorter than 10 and longer than 20. We assess generalisation by measuring the empirical coverage of predicted confidence intervals, as defined in section 3.2. Figure 5 shows results on the dataset with the scale factor 1.0.

We observe that while both models achieve reasonable calibration, the Restricted model exhibits slightly greater deviations from the nominal coverage, particularly for context lengths further from the training distribution. These results suggest that the probe generalises to novel context lengths, but training on a wider context ranges should be beneficial for a more robust generalisation.

### 5.2 APPLICABILITY TO REAL-WORLD DATA

Thus far, our analysis has focused on synthetic data. In this section, we evaluate whether our probing model can be trained successfully on real-world datasets, and how well predictions can generalise across different types of input series.

To assess this, we construct a dataset using time series from the Darts (Herzen et al., 2022) and Monash (Godahewa et al., 2021) collections. Following the same format as in our synthetic experiments, we generate LLM embeddings and samples for approximately 45,000 distinct sequences across 31 sub-datasets (e.g., US Births, Air Passengers). Furthermore, we also investigate an even stronger form of generalisation: from a model trained on synthetic data only to testing on real-world data. For this purpose, we train the following models:

Table 4: Coverage of the CI intervals on previously unseen testing inputs.

| Model | $\alpha$   50% | 90% | 95% |
|---|---|---|---|
| Real (all) | 48.8 ± 0.1 | 88.5 ± 0.1 | 94.3 ± 0.1 |
| Real (5 fold) | 43.4 ± 0.2 | 82.1 ± 0.2 | 89.4 ± 0.2 |
| Synth | 30.2 ± 0.3 | 67.7 ± 0.4 | 77.3 ± 0.4 |

- **Real (all):** Trained on a random 80% of all sequences across all sub-datasets. The remaining 20% is held out for testing.
- **Real (5 fold):** We partition the dataset into 5 folds such that, in each fold, one model is trained on 80% of the sub-datasets and evaluated on the remaining 20%. This ensures that each sub-dataset appears in the test fold of exactly one out of 5 models trained.
- **Synth:** A model trained on the combination of the 4 synthetic datasets with scales 1.0, 10.0, 1000.0 and 10000.0.

At test time, the above models face increasingly stronger distribution shifts. In terms of generalisation performance to previously unseen data distributions we can view the Real (all) model as an upper-bound baseline for Real (5 fold) and Synth.

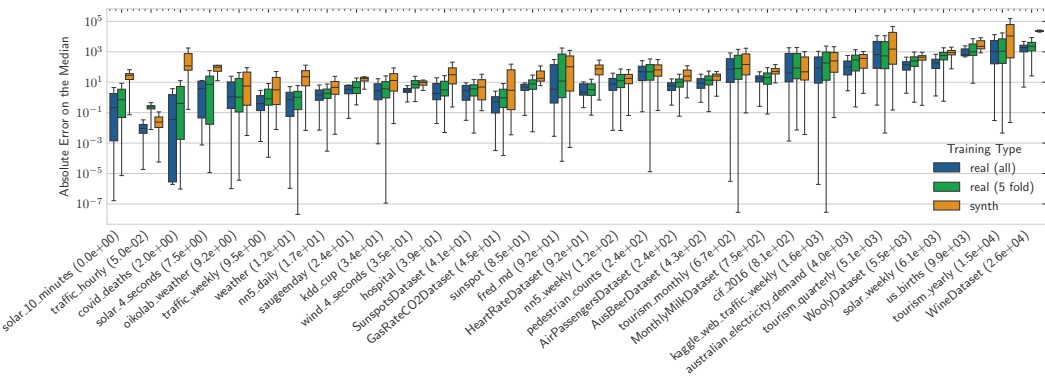

Figure 6: Absolute error on the median across different sub-dataset. Comparison of generalisation across models trained on different data.

In Table 4, we report the average coverage of the CI across all training types. We observe that the Real (all) model demonstrates good performance, with the empirical coverage of LLM samples closely matching the expected coverage. The Real (5 fold) model demonstrates a slight downgrade in performance. Interestingly, while the Synth model underperforms, it still demonstrates good generalisation for some of the sub-datasets as we can see on Figure 6. This figure shows the distribution of the absolute error of the predicted median vs. the median of LLM samples across all sub-datasets. The x-axis is sorted by increasing order of magnitude of the datasets defined by the average of the median of LLM samples. We note that the sub-datasets in our collection cover different ranges of values (with individual LLM samples varying in magnitude from $10^{-3}$ to $10^{13}$). We suspect that this is primarily why the probing model struggles to generalise across some datasets.

> ⚲ **Key Insights.** The probing model exhibit some, albeit limited, generalisation across context lengths. When applied to real-world datasets, the model achieves accurate empirical coverage and demonstrates partial transferability to unseen data distributions. Cross-dataset generalisation is possible, but challenged by large variation in scale and distribution.

# 6 DISCUSSION, LIMITATIONS AND FURTHER WORK

While most probing studies of LLMs focus on predicting categorical outputs in natural language tasks, we instead target numerical prediction, which is particularly challenging due to high variance in output magnitudes. To address this, we introduce novel probes that decompose the task into magnitude classification and scaled value regression. Our findings show that LLMs encode rich numerical information about their predictions *before* autoregressive decoding. Training lightweight probes on hidden states allows us to recover both point estimates (mean, median, greedy outputs) and uncertainty. This suggests that much of the LLM's "reasoning" over numerical outputs occurs during input processing, with autoregressive decoding primarily simply surfacing the predictions. Beyond offering insight into how LLMs handle regression, our results also open a practical path: enabling uncertainty-aware numerical prediction without the computational cost of repeated sampling.

Despite these promising results, several limitations remain. First, our approach requires access to internal model activations, even though it does not involve fine-tuning the LLM itself. Second, while our probing models exhibit some generalisation ability, they are still model-specific, requiring retraining for each architecture or tokenisation scheme. Third, training and evaluation requires approximating the LLM's predictive distribution via empirical sampling, which is an imperfect and computationally expensive proxy.

Future work could extend this framework to a broader range of structured data and prediction tasks, including univariate or multivariate regression and time-series forecasting, as well as multi-step ahead prediction tasks. While in this work we focused on quantifying the spread of the LLM distribution using quantile regression (which allowed us to obtain the estimates of the confidence intervals), alternative methods can be considered (e.g. Bayesian neural networks) which might offer a more fine-grained description of the distribution. Finally, motivated by our generalisation results, a key next step is the development of a universal probing model that can be applied off-the-shelf to a given LLM across diverse tasks and domains.

ACKNOWLEDGMENTS

We thank Claudio Fanconi for helpful suggestions on the design of the magnitude-factorised probing model. We also acknowledge the financial support of AstraZeneca (Julianna Piskorz) and Eedi (Katarzyna Kobalczyk). This work was supported by Azure sponsorship credits granted by Microsoft's AI for Good Research Lab.

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

## APPENDIX CONTENTS

## A  RELATED WORKS

**LLMs for Structured Data.** With the advances in the performance of LLMs, several methods have been proposed which utilise LLM's pre-training and prior knowledge to make predictions on structured data, namely *tabular data* (e.g. Requeima et al., 2024; Hegselmann et al., 2023; Shysheya et al., 2025; Vacareanu et al., 2024) and *time series data* (e.g. Gruver et al., 2024; Xue & Salim, 2023). These works show that LLMs can serve as competitive predictive models even *without task-specific fine-tuning*. This is particularly evident in the small sample regime, where large-scale pre-training, access to prior knowledge and ability to condition on textual information allows LLMs to match or outperform the performance of purpose-built regressors.

**Numerical Predictive Distributions of LLMs.** When used as regressors, LLMs can provide not only point estimates but also full predictive distributions, reflecting their stochastic nature. To elicit continuous distributions over numerical outputs, Gruver et al. (2024) and Requeima et al. (2024) propose an autoregressive approach that generates logit values over discretised numeric bins, which are then scaled to form a valid probability distribution. Access to such distributions is crucial for downstream tasks requiring uncertainty quantification, including decision-making under uncertainty and Bayesian optimisation. However, these methods are computationally intensive, as they require multiple sequential queries to the LLM to construct a single distribution (e.g., $p(123.4) = p(1)p(2|1)p(3|12)p(.|123)p(4|123.)$). This motivates us to explore alternative approaches to eliciting numerical predictive distributions from LLMs.

**Discrepancy between number generation and autoregression.** As next-token predictors, LLMs are not explicitly trained to understand the value of numbers. Due to their autoregressive nature, early tokens encode digits before key decisions like decimal placement (that determine a number's magnitude) are made. This can lead to surprisingly poor performance on simple numerical tasks (Yang et al., 2019; Akhtar et al., 2023; Zhou et al., 2024; Schwartz et al., 2024). To address these limitations, several works have proposed alternatives to standard autoregressive decoding for numerical predictions. For instance, Golkar et al. (2024) introduce a special `[NUM]` token, replaced post-hoc with a continuous value predicted by a learned regression head–though this requires retraining the model. Others (Singh & Strouse, 2024; Schwartz et al., 2024) investigate number-specific tokenizations to improve numerical accuracy of LLMs. In contrast, we ask whether one can bypass autoregressive decoding in *pre-trained* LLMs by directly reading out the predictive distribution from the internal representations.

**Probing and LLMs.** Probe classifiers, or simply probes, are models trained to uncover specific properties directly from the intermediate representations of neural models Alain & Bengio (2018). In the context of LLMs, a growing body of works from the area of mechanistic interpretability studied what kind of properties can be directly recovered from the internal representations of the LLMs (without the need for decoding), and where within the internal representations they are located (e.g. which layer). Amongst others, prior work has probed LLMs for properties such as factuality of the generated responses Obeso et al. (2025); Orgad et al. (2025); Azaria & Mitchell (2023), semantic entropy Kossen et al. (2024), toxicity Roy et al. (2023); Wen et al. (2023), text sentiment Palma et al. (2025) and even training order recency Krasheninnikov et al. (2025). While the previous works focus primarily on training probe classifiers, in this work we focus on *probe regressors*, proposing our magnitude-factorised probe as a method for handling targets with large range of continuous values (spanning several orders of magnitude). By focusing on continuous values, we uncover that the hidden states of the LLMs uncover much more fine-grained information than suggested by previous works (which probe primarily for binary–or binarised–properties).

Complementary findings from mechanistic interpretability suggest that, even in purely textual settings, LLM hidden states encode representations of tokens that the model is most likely to generate several tokens ahead (Pal et al., 2023; Lindsey et al., 2025; Belrose et al., 2025). In contrast to these works, we show that similar results hold also when using LLMs for numerical predictions, allowing to uncover not only the marginal distributions of tokens at each of the steps ahead, but also the properties of the entire numerical predictive distribution of the LLM (mean, median, quantiles).

**Probing numeracy in LLM embeddings.** A number of prior works give evidence that simple probing models can be used to learn numerical values encoded in the LLM embeddings. Wallace et al. (2019) has shown that the value of a number can be successfully decoded from its encoded word embedding (e.g., "71" $\rightarrow$ 71.0.). Stolfo et al. (2023) identified specific layers in LLMs that store numerical content, recoverable via simple linear probes, while Zhu et al. (2023) demonstrated that intervening on these layers alters generated outputs. More recently, Koloski et al. (2025) showed that LLM embeddings can serve as effective covariates in downstream regression models.

Taken together, these results support the hypothesis that it should be possible to train probes that efficiently approximate the numerical *predictive distribution* of the LLM, motivating our work.

## B DETAILS OF THE EXPERIMENTAL SETUP

### B.1 ASSETS AND LICENSING INFORMATION

The following existing assets were used to produce the experimental results:

- **Monash dataset** Godahewa et al. (2021)
- **Darts dataset** Herzen et al. (2022)
- **Llama-2-7B model** Touvron et al. (2023)
- **Llama-3-8B model** Grattafiori et al. (2024)
- **Phi-3.5-mini model** Abdin et al. (2024)
- **DeepSeek-R1-Distill-Llama-8B model** DeepSeek-AI (2025)

### B.2 COMPUTER INFRASTRUCTURE USED

**Hardware.** All experiments were conducted using 2 separate NC24rs_v3 instances and one NC80adis_H100_v5 instance on the Microsoft Azure cloud platform. These instances are a part of Azure's GPU-optimised virtual machine series, with their hardware specifications summarised in Table 5.

Generating the synthetic dataset for one scaling factor $D_{\text{scale}} \in \{1, 10, 1000, 10000\}$ took no more than 10h. Training one probe model took no more than 4h.

Table 5: Azure Virtual Machine Specifications

| Specification | NC24rs_v3 | NC80adis_H100_v5 |
|---|---|---|
| vCPUs | 24 | 80 |
| System Memory (GiB) | 448 | 640 |
| GPU Model | 4× NVIDIA Tesla V100 | 2× NVIDIA H100 NVL |
| GPU Memory (per GPU) | 16 GiB | 94 GiB |
| Total GPU Memory | 64 GiB | 188 GiB |
| GPU Architecture | Volta | Hopper |
| CUDA Version | 11.x | 12.x |
| CPU Model | Intel Xeon E5-2690 v4 | AMD EPYC Genoa |
| Local Storage | 2.9 TB | 7.1 TB |

## B.3 DETAILS OF THE DATASETS

### B.3.1 DETAILS OF THE SYNTHETIC TIME SERIES DATASET

We generate a synthetic dataset comprising time series derived from a family of parametric functions, each evaluated over a fixed domain and perturbed with controlled noise. The purpose is to simulate diverse temporal patterns, inducing varying levels of uncertainty in the LLM's predictions.

We use a set of base functions defined over the interval $x \in [0, 60]$, discretised into 120 equidistant points. The functions are summarised in Table 6. For each function and value of $a$, we generate a clean series $y = f(a \cdot x)$, and then apply:

- Additive Gaussian noise with variance $\sigma^2 \in \{0.0, 0.01, 0.05, 0.1\}$.
- Vertical scaling by $b \sim \mathcal{U}(0, D_{\text{scale}})$
- Vertical translation by $d \sim \mathcal{U}(-D_{\text{scale}}, D_{\text{scale}})$

From each transformed series, we sample 10 different subsequences for each length $n \in \{3, 5, 7, 10, 13, 15, 17, 20, 25, 30, 35, 40\}$, with random starting offsets. Each subsequence is used as a candidate training input $\mathbf{x}_i$. Inputs are serialised as floating-point strings with $p$ decimal places (we use $p = 4$ for $D_{\text{scale}} = 1.0$, $p = 3$ for $D_{\text{scale}} = 10.0$, $p = 2$ for $D_{\text{scale}} = 1000.0$ and $p = 1$ for $D_{\text{scale}} = 10000.0$). This results in 33600 generated raw time series for each value of $D_{\text{scale}}$.

**Concatenation, filtering, and final splits.** We then use the raw generated datasets to create four scale-specific datasets which are bounded in value and exhibit a balanced distribution across magnitude orders. Firstly, after querying the LLM to generate 100 predictive samples for each input sequence, we keep only those series for which 100 predictive samples were successfully generated. After this step, we combine all raw datasets and derive balanced subsets at each scale level by binning examples $\mathbf{x}_i$ according to the median value of the LLM prediction, $|y_i^{\text{median}}|$, using 8 logarithmic bins over $[10^{-3}, 10^4]$. We cap the number of samples in each bin at 12,000; bins may contain fewer samples if the initial data generation process does not yield a sufficient number of instances, particularly for bins corresponding to the smallest magnitudes. To avoid outlier values we also filter the data to maintain only the series $\mathbf{x}_i$ for which

$$|y_i^{\text{median}}|, |y_i^{\text{mean}}|, |y_i^{\text{greedy}}| < D_{\text{scale}}.$$

Thus, the final size of each scale-specific dataset is scale-dependent, with approximate sizes of 18k, 30k, 48k, and 60k for $D_{\text{scale}} \in \{1.0, 10.0, 1000.0, 10000.0\}$, respectively. Finally, for each dataset we create train/validation/test splits with proportions $0.8/0.1/0.1$.

### B.3.2 MONASH DATASET

- **Data Loading**: We use the data from the Monash dataset, preprocessed by Gruver et al. (2024) and available from `https://drive.google.com/file/d/1sKrpWbD3LvLQ_e5lWgX3wJqT50sTd1aZ/view?usp=sharing`. Each sub-dataset file contains tuples of the form `(train, test)`, which are concatenated to form complete univariate time series.

| Function name | Formula | $a$-range |
|---|:---:|:---:|
| `sin` | $\sin(x)$ | [0.5, 6.0] |
| `linear_sin` | $0.2 \cdot \sin(x) + \frac{x}{450}$ | [0.5, 6.0] |
| `sinc` | $\text{sinc}(x)$ | [0.05, 0.2] |
| `xsine` | $\frac{x-30}{50} \cdot \sin(x - 30)$ | [0.5, 1.3] |
| `beat` | $\sin(x) \cdot \sin\left(\frac{x}{2}\right)$ | [0.1, 6.0] |
| `gaussian_wave` | $e^{-\frac{(x-2)^2}{2}} \cdot \cos(10\pi(x - 2))$ | [0.01, 0.1] |
| `random` | $\mathcal{U}(-1, 1)$ | [0.0, 1.0] |

Table 6: Functions used to generate time series data, their mathematical forms, and the range of the time-scaling parameter $a$.

- **Resampling**: To ensure computational tractability, each series is subsampled (via strided slicing) to contain at most 1000 time steps.

- **Series Selection**: For each dataset, a maximum of 50 time series are selected at random to control the number of examples used during training.

- **Subsequence Generation**: From each selected series, we extract multiple training subsequences of varying lengths $n \in \{3, 5, 7, 10, 13, 15, 17, 20, 25, 30, 35, 40\}$. For each length, we generate up to 10 training subsequences, sampled at different offsets.

### B.3.3 DARTS DATASET

- **Data Loading**: We use the data from the Darts dataset, available from the `darts` python package. We use the following sub-datasets: AirPassengersDataset, AusBeerDataset, GasRateCO2Dataset, MonthlyMilkDataset, SunspotsDataset, WineDataset, WoolyDataset, HeartRateDataset.

- **Resampling**: To ensure computational tractability, the series for the datasets Sunspots-Dataset and HeartRateDataset are subsampled (via strided slicing).

- **Series Selection**: For each dataset, all available time series are selected.

- **Subsequence Generation**: From each selected series, we extract multiple training subsequences of varying lengths $n \in \{3, 5, 7, 10, 13, 15, 17, 20, 25, 30, 35, 40\}$. For each length, we generate up to 10 training subsequences, sampled at different offsets.

### B.3.4 LLM GENERATION SETTINGS

We generate the LLM hidden states from LLMs available through the `huggingface` library. For each of the input time series, we obtain 100 samples from the LLM, generated autoregressively, as well as the greedy generation. During generation for Llama-2-7b, as its tokeniser encodes each digit separately, we narrow down the generated tokens to just the digits, decimal point and $+/-$ signs. For obtaining the random samples, we use `temperature=1.0` and `top_p=0.95`. We exclude from the final dataset samples for which generation failed at least once (i.e. the obtained generation was not a valid number), such that each time series in the final dataset has exactly 100 valid LLM samples.

### B.3.5 TRAIN-VALIDATION-TEST SPLIT

Before training, we split each of the datasets in $80\%$ training dataset, $10\%$ validation dataset and $10\%$ test dataset. Unless otherwise stated (in the generalisation experiments), these splits are random. We do not apply any scaling or transformation to either the LLM embeddings (which are inputs to our model) or the outputs.

### B.4 DETAILS OF THE PROBING MODELS

Our magnitude-factorised regression models, used both for the purpose of point prediction and for the purpose of quantile regression have the hyperparameters as reported in Table 7 and Table 8. We train

the model using the ADAM optimiser. Details of the implementation can be found in our codebase, available at `https://github.com/kasia-kobalczyk/guess_llm.git`.

| Hyperparameter | Description | Default Value |
|---|---|---|
| min_mag | Minimum exponent for base-10 magnitude scaling (as used by $f_{\text{order}}$) | $-3$ |
| max_mag | Maximum exponent for base-10 magnitude scaling | $\log_{10}(D_{\text{scale}})$ |
| alpha | Weight for classification loss component | 100.0 |
| beta | Weight for regression loss component | 50.0 |
| K | Top-$K$ exponents taken into consideration (see Equation 4) | 3 |
| hidden_layers | Number of hidden layers in feature extractor | 1 |
| hidden_dim | Dimensionality of hidden feature representation | 512 |
| activation_func | Activation function used in the MLP | GELU |
| hidden_states_list | A list of the hidden states $\mathcal{H}$ to use as input | $[25, \ldots, 32]$ |

Table 7: Model-specific hyperparameters for the magnitude-factorised models.

| Hyperparameter | Description | Default Value |
|---|---|---|
| learning_rate | Learning rate for the optimiser | $10^{-5}$ |
| weight_decay | L2 regularization weight | 1.0 |
| scheduler_step_size | Learning rate scheduler step size | 100 |
| scheduler_gamma | Learning rate scheduler gamma | 0.5 |
| batch_size | Number of samples per training batch | 1024 |
| max_epochs | Number of training epochs | 600 |
| patience | Patience for the early stopping | 200 |

Table 8: Optimiser and training-related hyperparameters.

## C  ADDITIONAL EXPERIMENTAL RESULTS

### C.1  COMPARISON AGAINST VANILLA MLP PROBE

In Table 9, we compare our magnitude-factorised probe with a vanilla MLP trained on the same hidden representations. Both models use a single hidden layer of size 512 and a learning rate of $10^{-5}$. Evaluation is conducted on Llama-2 using the synthetic time series dataset with scale $D_{\text{scale}} = 1.0$, which provides the narrowest range of magnitudes among those we study. Even in this relatively simple setting, the magnitude-factorised probe achieves substantial improvements over the baseline, reducing MSE by 41%, 33%, and 42% for greedy, mean, and median predictions respectively. These results highlight the importance of probe design: a negative finding with one architecture does not necessarily imply that information is absent from the LLM's hidden states—it may simply reflect the limitations of the probing model.

Table 9: MSE of the values predicted with the magnitude-factorised (MFP) model vs. a vanilla MLP.

| predicted value | MFP probe (ours) | MLP probe (log-scaling) | MLP probe (no log-scaling) |
|---|---|---|---|
| greedy | 0.0150 | 0.0400 | 0.0255 |
| mean | 0.0061 | 0.0091 | 0.0092 |
| median | 0.0058 | 0.0101 | 0.0100 |

## C.2    RESULTS WITH OTHER LLMs

We provide results for the key experiments in the main paper with more LLMs. As the tokenisers of models outside of the Llama-2 family do not encode digits separately, we *do not* narrow down the generated tokens during decoding. For obtaining the random samples, we use `temperature=1.0` and `top_p=0.95`. We perform repeated sampling until for each time series, we obtain 100 LLM samples $y_i^j \sim p_{\text{LLM}}(\cdot|\mathbf{x}_i)$. For Llama-3.2-3B, as the model contains less layers, we also concatenate the representations from across the last 8 layers, which amounts to layers $[21, \ldots, 28]$.

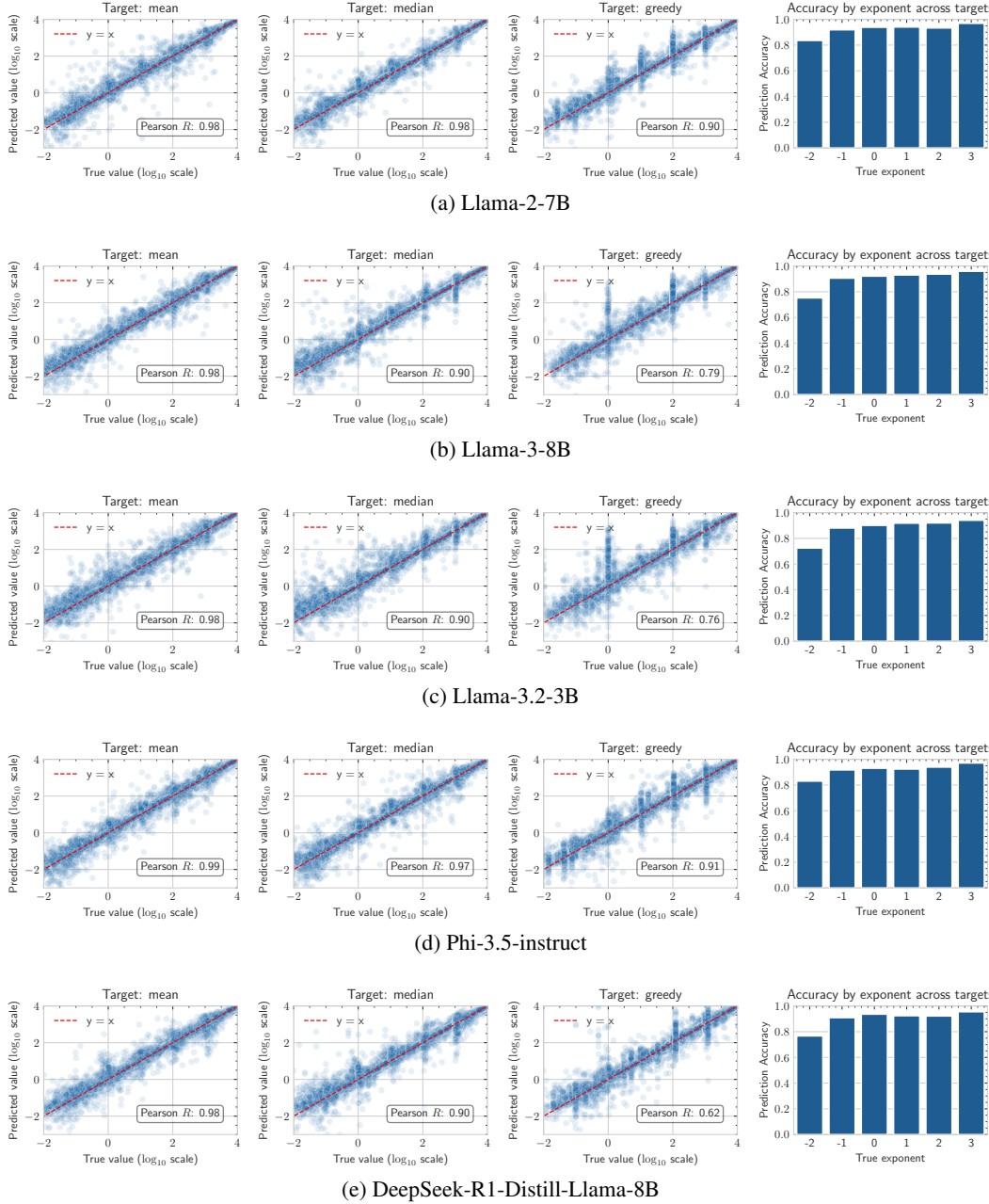

Figure 7: Predicted vs. sample mean, median and greedy prediction (on $\log_{10}$ scale). Results analogous to Figure 2.

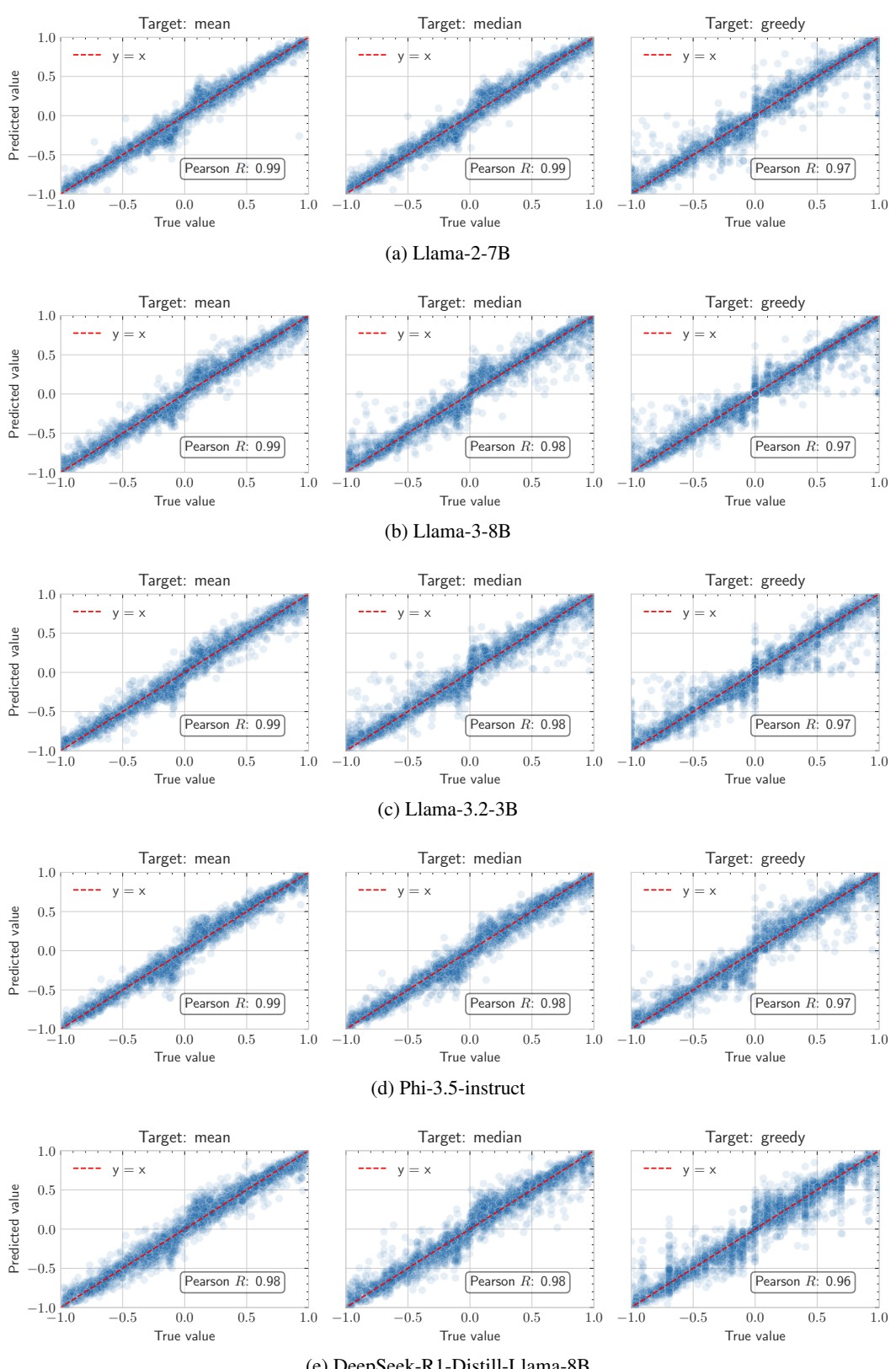

Figure 8: Predicted vs. sample mean, median and greedy prediction on the dataset with scale $D_{\text{scale}} = 1.0$.

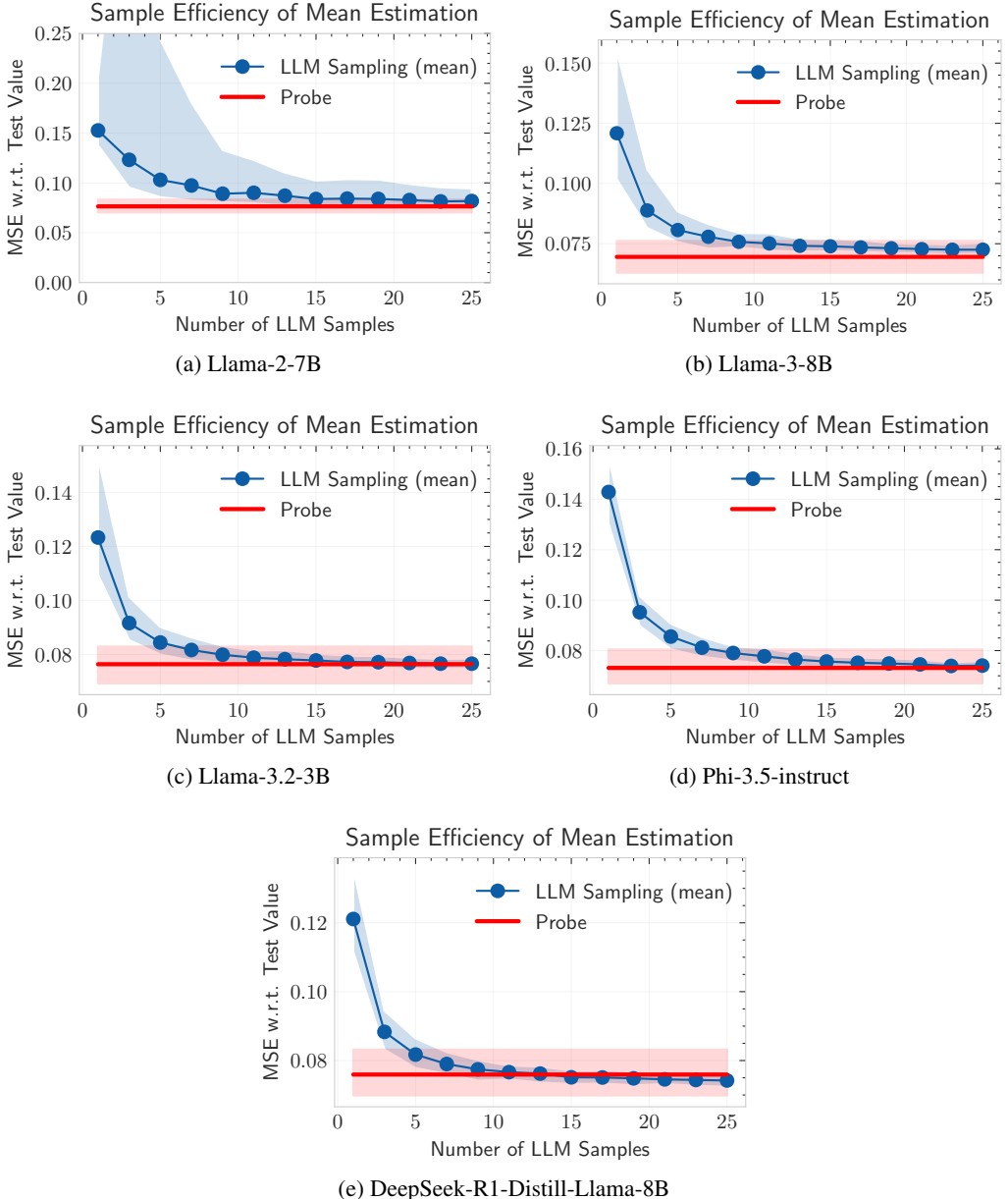

Figure 9: Sample efficiency on the task of one step ahead predictions. Results analogous to Figure 4. The shaded regions mark 95% confidence sets, computed using 100 bootstrap samples. We use $D_{\text{scale}} = 1.0$.

Table 10: MSE for the predictions on the dataset with scale $D_{\text{scale}} = 1.0$, reported for all models. Results analogous to Table 1.

(a) Llama-2-7B

| $y_i^*$, target | $\hat{y}_i^*$ (ours) | $\bar{\mathbf{x}}$ | $\bar{\mathbf{x}}_i$ | $x_{i,n}$ |
|---|---|---|---|---|
| $* = $ mean | 0.006 | 0.256 | 0.035 | 0.085 |
| $* = $ median | 0.006 | 0.260 | 0.041 | 0.087 |
| $* = $ greedy | 0.015 | 0.273 | 0.065 | 0.109 |

(b) Llama-3-8B

| $y_i^*$, target | $\hat{y}_i^*$ (ours) | $\bar{\mathbf{x}}$ | $\bar{\mathbf{x}}_i$ | $x_{i,n}$ |
|---|---|---|---|---|
| $* = $ mean | 0.007 | 0.253 | 0.047 | 0.093 |
| $* = $ median | 0.012 | 0.264 | 0.061 | 0.106 |
| $* = $ greedy | 0.016 | 0.255 | 0.072 | 0.122 |

(c) Llama-3.2-3B

| $y_i^*$, target | $\hat{y}_i^*$ (ours) | $\bar{\mathbf{x}}$ | $\bar{\mathbf{x}}_i$ | $x_{i,n}$ |
|---|---|---|---|---|
| $* = $ mean | 0.007 | 0.260 | 0.0484 | 0.092 |
| $* = $ median | 0.013 | 0.271 | 0.062 | 0.104 |
| $* = $ greedy | 0.018 | 0.267 | 0.075 | 0.119 |

(d) Phi-3.5-mini-instruct

| $y_i^*$, target | $\hat{y}_i^*$ (ours) | $\bar{\mathbf{x}}$ | $\bar{\mathbf{x}}_i$ | $x_{i,n}$ |
|---|---|---|---|---|
| $* = $ mean | 0.008 | 0.248 | 0.042 | 0.100 |
| $* = $ median | 0.012 | 0.252 | 0.047 | 0.104 |
| $* = $ greedy | 0.020 | 0.270 | 0.060 | 0.113 |

(e) DeepSeek-R1-Distill-Llama-8B

| $y_i^*$, target | $\hat{y}_i^*$ (ours) | $\bar{\mathbf{x}}$ | $\bar{\mathbf{x}}_i$ | $x_{i,n}$ |
|---|---|---|---|---|
| $* = $ mean | 0.009 | 0.247 | 0.045 | 0.110 |
| $* = $ median | 0.013 | 0.253 | 0.056 | 0.120 |
| $* = $ greedy | 0.020 | 0.264 | 0.069 | 0.135 |

Table 11: Coverage of the CI for all models. Results analogous to Table 2.

(a) Llama-2-7B

| $\alpha$ dataset | 50% | 90% | 95% |
|---|---|---|---|
| 1.0 | $52.0 \pm 0.4$ | $90.9 \pm 0.3$ | $95.5 \pm 0.2$ |
| 10.0 | $52.7 \pm 0.5$ | $91.3 \pm 0.3$ | $96.1 \pm 0.2$ |
| 1000.0 | $51.4 \pm 0.3$ | $90.7 \pm 0.3$ | $95.7 \pm 0.2$ |
| 10000.0 | $48.2 \pm 0.3$ | $90.5 \pm 0.2$ | $95.4 \pm 0.2$ |

(b) Llama-3-8B

| $\alpha$ dataset | 50% | 90% | 95% |
|---|---|---|---|
| 1.0 | $54.4 \pm 0.6$ | $90.4 \pm 0.4$ | $94.7 \pm 0.3$ |
| 10.0 | $53.8 \pm 0.6$ | $91.8 \pm 0.4$ | $96.3 \pm 0.2$ |
| 1000.0 | $51.7 \pm 0.4$ | $91.2 \pm 0.3$ | $96.0 \pm 0.2$ |
| 10000.0 | $50.8 \pm 0.4$ | $91.1 \pm 0.3$ | $95.3 \pm 0.2$ |

(c) Phi-3.5-mini-instruct

| $\alpha$ dataset | 50% | 90% | 95% |
|---|---|---|---|
| 1.0 | $49.9 \pm 0.5$ | $89.3 \pm 0.4$ | $94.3 \pm 0.3$ |
| 10.0 | $50.9 \pm 0.5$ | $89.3 \pm 0.4$ | $95.1 \pm 0.3$ |
| 1000.0 | $47.6 \pm 0.4$ | $88.0 \pm 0.3$ | $93.3 \pm 0.3$ |
| 10000.0 | $47.3 \pm 0.3$ | $87.3 \pm 0.3$ | $93.8 \pm 0.2$ |

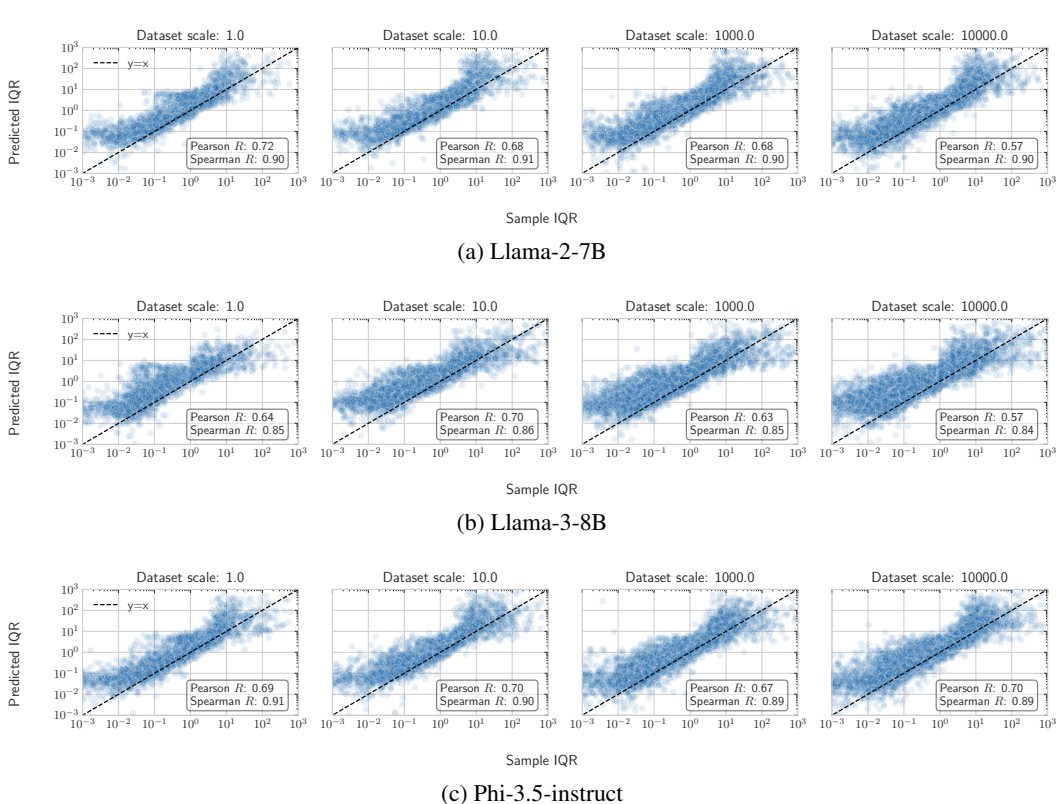

Figure 10: Predicted IQR vs. Sample IQR (median adjusted). Results analogous to Figure 3.

### C.3 COMPUTATIONAL COSTS AND INFERENCE TIME

Crucial for understanding the validity of our approach is the comparison of the inference cost of the proposed probe vs. autoregressive sampling. Ignoring the cost of encoding the input time series with the LLM (as this cost is shared by both methods) we note that decoding each token with the 7B-parameter LLMs (such as Llama-2) requires $14B$ FLOPS, which means that, for a model with digit-by-digit tokenization obtaining $n$ samples of a 5-digit number requires $n \times 70B$ FLOPS. In comparison, using our magnitude-factorised probe (which we assume uses 8 concatenated layers of the LLM and thus has input of size $8 \times 4,096 = 32,768$, 1 hidden layer of dimension $512$, and a maximum of $k = 9$ separate heads corresponding to each of the magnitude bins) incurs the constant cost of $32,768 \times 512 + 512 \times 9 \approx 17M$ FLOPS for the magnitude classification model and $32,768 \times 512 + 512 \times 1 \approx 17M$ FLOPS for the regression model, so $34M$ parameters in total per one statistic. If we wanted to predict, say, 7 different quantile values, we would need 7 models of the same size, requiring $234M$ FLOPS. Thus, the required number of computations for using the probe vs. generating even 1 LLM sample is far lower.

We further validate these estimates by comparing the times needed to generate $n$ samples from the LLM vs. the time needed to obtain a single median estimate from our probe. We run the experiments using the Llama-2-7B model, averaging the results over 100 random time series samples consisting of 20 data points each. We run all the timing experiments on one instance of a H100 GPU. In this setting, inference with a trained probe model (including the process of obtaining the hidden representation of the LLM) takes $0.034 \pm 0.006$s. We include the times required for obtaining $n$ LLM samples in Table 12. These results confirm that even generating a single sample via autoregression is roughly $47\times$ slower than running the full inference pipeline with our probe.

Table 12: Average time to generate $n$ LLM samples.

| $n$ samples | 1 | 5 | 10 | 20 | 50 | 100 |
|---|---|---|---|---|---|---|
| time (s) | $1.59 \pm 0.08$ | $1.62 \pm 0.07$ | $1.71 \pm 0.07$ | $1.83 \pm 0.07$ | $2.35 \pm 0.08$ | $3.28 \pm 0.08$ |

### C.4 LAYER ABLATION STUDY

In this section, we ablate the choice of the layers $\mathcal{H}$ used by our probing models by training on individual layers of the LLM, letting $\mathcal{H} = \{\ell\}$ for each $\ell \in [16, \ldots, 32]$. Our results provide further insights into how information about the LLM's numerical predictive distribution is spread across the layers.

Figure 11 shows the results for the probing models from section 2. For the target of the greedy LLM prediction, we see that the most relevant information is contained in the last layers of the LLM, although generally the variations between the layers are not that significant. We further observe that concatenating information from across different layers leads to significant performance improvements. For the mean target, which is predicted with a higher accuracy, we observe that information relevant for predicting the magnitude and the scaled value of the target is distributed less uniformly, concentrating in the 30th layer.

Table 12 shows analogous results for the quantile regression model from section 3.2. The observed pattern for median prediction is similar to the one of mean prediction from Figure 11. In terms of the uncertainty information, we find it to be more uniformly encoded across the layers and again that concatenating information across several layers leads to an improved probing performance.

### C.5 MEAN ABSOLUTE LOSS PER QUANTILE

In addition to the results presented in Section 3, we report the mean absolute error between the the empirical quantile $\tilde{q}^s$ and the predicted quantile $\hat{q}^s$ for all quantile levels. Results are shown in Table 13. We observe that the absolute errors are higher for quantile levels further from the median.

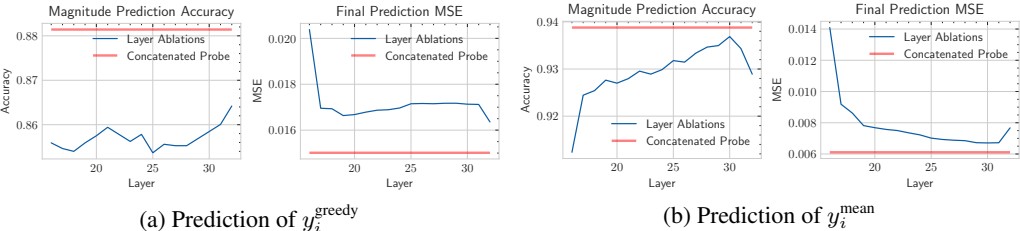

(a) Prediction of $y_i^{\text{greedy}}$        (b) Prediction of $y_i^{\text{mean}}$

Figure 11: *Layer ablation for Llama-2-7B on the probing model from section 2.*

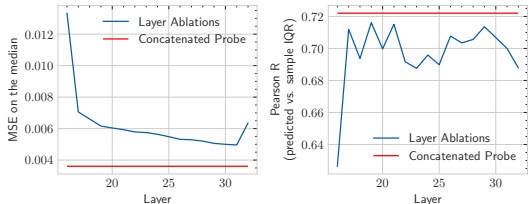

Figure 12: *Layer ablation for Llama-2-7B on the quantile probing model from section 3.* MSE on the median (left, lower is better) and the Pearson R between predicted IQR and sample IQR (right, higher is better).

Table 13: *Quantile regression – mean absolute error per quantile.* Values represent the mean absolute error between the empirical quantile $\tilde{q}^s$ and the predicted quantile $\hat{q}^s$ for all quantile levels.

| quantile level $\tau^s$ dataset | 0.025 | 0.05 | 0.25 | 0.5 | 0.75 | 0.95 | 0.97 |
|---|---|---|---|---|---|---|---|
| 1.0 | 0.58 | 0.40 | 0.16 | 0.05 | 0.15 | 0.40 | 0.60 |
| 10.0 | 3.12 | 2.16 | 0.45 | 0.28 | 0.49 | 1.91 | 3.26 |
| 1000.0 | 111.16 | 61.54 | 14.83 | 12.10 | 14.30 | 46.04 | 115.71 |
| 10000.0 | 843.11 | 380.06 | 119.80 | 100.41 | 118.90 | 483.33 | 1172.27 |

