# OpenReview forum: "Eliciting Numerical Predictive Distributions of LLMs Without Auto-Regression"
_ICLR.cc/2026/Conference — ICLR 2026 Poster_

### Official Review · Reviewer_8jM1 · 2025-10-24

**Soundness:** 3
**Presentation:** 4
**Contribution:** 2
**Rating:** 4
**Confidence:** 4

**Summary:**

In an empirical paper, authors target numerical prediction in LLMs by decomposing prediction into magnitude classification and scaled value regression. The key contribution is learning the magnitude and a normalized value separately, and using the predictions before LLM token generation.

While it is hinted that the idea enables uncertainty-aware numerical
prediction without repeated sampling, evaluation is limited to maximum Pinball loss and IQR prediction in generated data.

While the methods empirically work well, we gain no insight regarding the LLM representation behind.

The paper is well written and source code is provided anonymously.

While the paper presents convincing empirical results for the applicability of the decomposed numerical prediction learning in LLMs, I miss theoretical or architectural explanations. Also, the uncertainty considerations are shomewhat limited.

**Strengths:**

+ New contribution towards improved numeric prediction capabilities of LLMs
+ Very well written paper
+ Source code is available

**Weaknesses:**

- No theoretical and/or architectural explanation of how the LLMs represent the numeric range and scaled value
- The uncertainty part could be strengthened
- Comparison with methods that learn the distribution of regression problems (e.g. Bayesian NN, Mixture Density Networks), while not absolutely necessary, could make the contribution more valuable

**Questions:**

The discussion of Table 1 could be more elaborate. Do I understand well that the LLM raw output is much worse than the mean or median of several of its outputs? I see no explanation, not even in the Appendix.

3.1 pinball loss: why the maximum over quantiles? Why not the integral, as in CRPS [Alexander Jordan, Fabian Kruger, and Sebastian Lerch. Evaluating probabilistic forecasts with scoringrules. Journal of Statistical Software, 90(12):1–37, 2019, Diane Bouchacourt, Pawan K Mudigonda, and Sebastian Nowozin. Disco nets: Dissimilarity coefficients networks, Neurips'16]?

In the training procedure of eqs (6-7), how can you handle if a multimodal distribution has modes of different order of magnitude? Isn't eq (6) too restrictive?

While the main goal is certainly not to provide the best model for distribution learnting, could you compare other methods that learn the distribution, e.g. [Shengyang Sun, Changyou Chen, and Lawrence Carin. Learning Structured Weight Uncertainty in Bayesian Neural Networks. AISTATS'17].

Fig. 6 only shows MAE, it would be interesting to see e.g. Pinball Loss, compared to non-LLM models for learning distributions.

---

> ### Author Response · Authors · 2025-11-24
> **Rebuttal (Part 1/3)**
>
> We thank the reviewer for their thoughtful and constructive feedback. Below we address all your comments point by point.
>
> ## **Weaknesses**
>
> > **W1.** No theoretical and/or architectural explanation of how the LLMs represent the numeric range and scaled value.
> >
>
> Thank you for this question, we also find it intriguing to see how the LLM encodes information necessary to recover the future predictions. To provide insights in this direction, **Appendix C.4** of our submission includes the results of an ablation study aiming to identify which layers contain most information about both the point predictions (sample mean, greedy prediction) and the uncertainty (quantiles). Results in Figure 11 demonstrate that the point predictions (particularly the mean) are mostly encoded in the last layer of the LLM. There is no significant difference between where the magnitude of the number and its scaled value are encoded. In turn, uncertainty (IQR) is encoded more uniformly across the layers.
>
> With our experiments, we have also found that predicting the mean or the median of the predictive distribution with the probing method can be done with a significantly lower error than recovering the greedy prediction (Table 1). We hypothesise this to be the case because the greedily decoded output is not strongly represented in the hidden states, unlike central distributional statistics such as the mean or median. The greedy prediction is a byproduct of the decoding procedure rather than a meaningful property of the underlying predictive distribution. We find this to be an additional interesting observation shedding insight into what properties are encoded in the LLM’s hidden states.
>
> > **W2.** The uncertainty part could be strengthened
> >
>
> Thank you for raising the point on strengthening the uncertainty results. We have introduced small architectural improvements to the quantile regression model, making it more closely aligned with the architecture presented in section 2 of our work. In summary, in the updated version of the method, each predictive head for each quantile value adopts the same conditional architecture as the probe for single value prediction in section 2. We have updated the manuscript to directly reflect these changes.
>
> This modification leads to noticeably improved IQR predictions: in the revised Figure 3, the scatter plots exhibit a smoother concentration around the diagonal $y=x$. This improvement is also reflected in calibration performance. In the original submission, empirical coverage was:
>
> | $\alpha$ [%] | **50** | **90** | **95** |
> | --- | --- | --- | --- |
> | 1.0 | 47.3 | 88.2 | 93.3 |
> | 10.0 | 52.3 | 89.5 | 93.8 |
> | 1000.0 | 48.9 | 87.2 | 92.7 |
> | 10000.0 | 46.5 | 86.0 | 91.3 |
>
> In the updated version, calibration improves across all settings:
>
> | $\alpha$ [%] | **50** | **90** | **95** |
> | --- | --- | --- | --- |
> | 1.0 | 52.0 | 90.9 | 95.5 |
> | 10.0 | 52.7 | 91.3 | 96.1 |
> | 1000.0 | 51.4 | 90.7 | 95.7 |
> | 10000.0 | 48.2 | 90.5 | 95.4 |
>
> The empirical coverage now deviates from the expected level by at most 2.7 percentage points (previously up to 4pp).
>
> Finally, we observe an improved generalisation from a probe trained on synthetic data to testing on real-world data (Table 2, Synth, expected 90% was 56.4%, is 67.7%, expected 95% → was 67.4% is 77.3%).
>
> > **W3.** Comparison with methods that learn the distribution of regression problems (e.g. Bayesian NN, Mixture Density Networks), while not absolutely necessary, could make the contribution more valuable
> >
>
> We appreciate the reviewer’s suggestion. While there are indeed alternative approaches for modelling predictive distributions (e.g., Bayesian neural networks or mixture density networks), our goal in Section 3 of the paper is to answer a specific scientific question: whether information about the uncertainty of the LLM’s predictive distribution is encoded in its hidden states and can be recovered by a lightweight probing model. For this purpose, the magnitude-factorised quantile regression approach proved effective, allowing us to directly test recoverability of distributional information.
>
> We also note that applying alternatives such as BNNs or MDNs in our setting would require addressing the large range of the target values—a challenge that our magnitude-factorised formulation explicitly resolves by separating magnitude prediction from scaled value regression. Without such factorisation, these alternative methods are likely to face stability and calibration issues—as we observed to be the case when training simple MLPs for the prediction of single scalar statistics of the LLM’s predictive distribution (see Appendix C.1 for an experimental comparison against simple probing approaches).
>
> Nevertheless, we agree that exploring alternative probabilistic probing architectures could be an interesting direction for future work, which we have now acknowledged in the discussion section.

---

> ### Author Response · Authors · 2025-11-24
> **Rebuttal (Part 2/3)**
>
> ## **Questions**
>
> > **Q1**. The discussion of Table 1 could be more elaborate. Do I understand well that the LLM raw output is much worse than the mean or median of several of its outputs? I see no explanation, not even in the Appendix.
> >
>
> In Section 1, Table 1 we evaluate how good our probe is at predicting the corresponding statistic of the LLM’s predictive distribution on a held out test set. For example, in the row with target set to ‘mean', we present the MSE calculated as $\frac{1}{N}\sum\_{i=1}^N (y\_i^{mean} - \hat{y}\_i^{mean})^2$, where $y\_i^{mean}$ is calculated as the empirical mean of 100 samples from the LLM's predictive distribution: $y\_i^{mean} = \frac{1}{100}\sum\_{j=1}^{100}y_i^j, \; y\_i^j \sim p_{LLM}(\cdot|\mathbf{x}\_i)$. In turn, $\hat{y}\_i^{mean}$ is the prediction of the probing model trained to predict $y_i^{mean}$, with the LLM's hidden states $\mathbf{e}\_i$ as the input. To put our results on a relative scale, we compare against three simple baselines based on the statistics of the training data or the input time series $\mathbf{x}\_i$ itself. In particular, we emphasise that in Table 1 we do not compare how good the LLM’s outputs are at predicting the next ground-truth value of the time series $x\_{i, n+1}$ (for this, we refer to Section 4.1, Table 3).
>
> Regarding the results, we note that the accuracy of our probe is significantly worse when predicting $y^{greedy}$ than predicting $y^{mean}$ or $y^{median}$. As we explain in lines 210-213, we believe that this is because the greedy prediction is not an explicit property of the LLM’s predictive distribution (such as mean, median or the mode), but rather a by-product of the autoregressive decoding process. In consequence this information is not as cleanly encoded in the LLMs hidden states making it harder to recover with our probing model.
>
> With respect to the accuracy of the greedy, mean or median prediction on the one step ahead prediction task, Table 3 shows that the mean and the median of the LLM’s predictive distribution results in the lowest MSE. This agrees with the results of prior work (e.g. Requeima et al., 2024).
>
> Given these two observations we can conclude that recovering the greedy prediction from the LLM’s hidden states is both less reliable and less useful than recovering the central distributional statistics.
>
> > **Q2**. 3.1 pinball loss: why the maximum over quantiles? Why not the integral, as in CRPS [Alexander Jordan, Fabian Kruger, and Sebastian Lerch. Evaluating probabilistic forecasts with scoringrules. Journal of Statistical Software, 90(12):1–37, 2019, Diane Bouchacourt, Pawan K Mudigonda, and Sebastian Nowozin. Disco nets: Dissimilarity coefficients networks, Neurips'16]?
> >
>
> Regarding Equation 4: to avoid misunderstandings, we note that our formulation of the PinballLoss using the $\max$ operator is mathematically equivalent to the alternative definition of the pinball loss, given by:
>
> $\textrm{PinballLoss}(\tau, \hat{q}, y) = (\tau - \mathbb{I}\{y < \hat{q}\})(y-\hat{q})$
>
> In this work, we focus on recovering the quantiles of the LLM’s predictive distribution, as a way of characterising the underlying distribution. This choice allows us to (i) avoid making parametric assumptions on the LLM’s distribution (e.g. Gaussianity), thus allowing to effectively model the predictive distribution in the case when it is multimodal or non-symmetric; (ii) recover the confidence intervals for each input, which we assume are most important for downstream applications. Modelling the entire CDF $F_{\textrm{LLM}}(\cdot | x_{1:n})$ using a loss function such as the CRPS could be an alternative strategy, if a higher level of granularity than that provided by the quantiles was required. However, as a strictly more difficult problem, we expect training a probing model with CRPS to introduce more challenges in optimisation. Given this, we settled on employing the quantile regression approach which proved to be effective and sufficient for addressing the key questions posed in this paper.

---

> ### Author Response · Authors · 2025-11-24
> **Rebuttal (Part 3/3)**
>
> > **Q4**. While the main goal is certainly not to provide the best model for distribution learnting, could you compare other methods that learn the distribution, e.g. [Shengyang Sun, Changyou Chen, and Lawrence Carin. Learning Structured Weight Uncertainty in Bayesian Neural Networks. AISTATS'17].
> >
>
> See our answer to W3.
>
> > **Q5**. Fig. 6 only shows MAE, it would be interesting to see e.g. Pinball Loss, compared to non-LLM models for learning distributions.
> >
>
> We would like to clarify that in our paper, *we do not use the LLM to learn distributions.* Instead, our aim is to predict the statistics *of the LLM’s predictive distribution* $p_{\text{LLM}}(x_{n+1} \vert x_{1:n})$ from its hidden states (concretely, from the hidden representation of the last input token) using simple MLP-based models. It is unclear to us what the comparison to non-LLM models should entail and we would appreciate a clarification on what additional comparisons the reviewer would like us to present.
>
> ---
>
> Thank you again for all your comments which helped us improve the quality of our submission. We hope that our response has addressed your concerns and that the proposed revisions justify a more favourable assessment of our work. If the reviewer has any outstanding questions, we are happy to engage in further discussions.

---

### Official Review · Reviewer_hDqn · 2025-10-28

**Soundness:** 3
**Presentation:** 3
**Contribution:** 2
**Rating:** 6
**Confidence:** 2

**Summary:**

The authors explore the ability to extract the predictions of an LLM dependent on its hidden states of its final layers using a variety of methods. For their main experiment, they adopt a mantissa+exponent floating point style method of calculating the regression model, with the mantissa trained as a regression problem and the exponent trained as a classification problem. Only the exponent is trained first, then the mantissa is used to fine-tune the results. They then compare this against the point estimates of the greedy, median, and greedy methods of the LLM. They additionally train a quantile-regression probe using pinball loss to quantify uncertainty.

**Strengths:**

Strengths
This is an interesting dive into whether both 1. point estimates and 2. uncertainty can be recovered from the LLM's hidden state.

They show advantages against standard LLM sample-based prediction in Figure 4, along with improved results over GP in the tables for the time-series regression task.

They also demonstrate some generalization properties, which can be instrumental when dealing with distribution shift.

**Weaknesses:**

Though the author's main goal seems to be moving towards sidestepping autoregressive generation, they only explore one-step prediction in the current results. Additionally, for some of the main other results, they calculate statistics and last step predictions, it may be worth it to show average MSE across time rather than MSE against the average in time.

Currently, the experiments seem to be all done on synthetic datasets. While this is useful for the uncertainty metrics, having some error-based analysis on some standard real-world time series datasets could strengthen the paper.

**Questions:**

If LLMs generate results autoregressively and the model does not work as well for the greedy approach, does this mean that it would still be hard to capture the results of the LLMs purely from the hidden states?

How much contribution does the floating-point formulation actually bring for the probe, compared to just predicting the value directly? If it is significant, then how would this method compare to using it against a standard regressor (without LLM) with the same floating-point formulation?

Why is the LLM sample error so high in the beginning for Figure 4?

---

> ### Author Response · Authors · 2025-11-24
> **Rebuttal (Part 1/2)**
>
> We thank the reviewer for their thoughtful and constructive feedback. We appreciate the recognition of our contribution, and are grateful for the reviewer’s suggestions which helped us identify areas for clarification. Below we address all your comments point by point.
>
> ---
>
> ## **Weaknesses**
>
> > **W1**. Though the author's main goal seems to be moving towards sidestepping autoregressive generation, they only explore one-step prediction in the current results. Additionally, for some of the main other results, they calculate statistics and last step predictions, it may be worth it to show average MSE across time rather than MSE against the average in time.
> >
>
> It is correct that our probe predicts only the next-step distribution $p(x_{n+1} \mid x_{1:n})$. This is fully consistent with standard practice in time-series forecasting and autoregressive modelling: many widely used models (e.g., ARIMA, state-space models) are trained to estimate the **one-step-ahead** predictive distribution, with multi-step forecasting obtained via recursive application (predict $\hat{x}\_{n+1}$, append it to the input, predict $\hat{x}_{n+2}$, etc.). We do not claim that our method avoids autoregression over time. Rather, our contribution is that it avoids **autoregression at the token level of the next number**.
>
> Let us clarify that the focus of this work is to bypass **token-level** autoregressive generation of *individual numbers*. As noted in lines 102-105, modern LLM tokenizers decompose numeric values into multiple tokens (e.g. $12345.67$ is tokenized into [123][45][.][67] by Llama-3 and into [1][2][3][4][5][.][6][7] by Llama-2). Generating a single numerical value therefore requires several passes through the LLM, creating substantial computational overhead. Our results show that this overhead can be avoided by decoding numerical predictions directly from hidden states.
>
> We agree that extending our framework to predict distributions over multiple future values is an interesting direction for future work.  However, this pursuit faces non-trivial design choices which are dependent on the downstream application: should the probe target just the marginal distributions $p(x_{n+k} | x_1, \dots, x_{n})$ for $k \geq 1$, or also capture dependencies on the intermediate predictions $x_{n+1}, \dots, x_{n+k-1}$?  The present contribution demonstrates that next-value predictive distributions are recoverable from internal representations using a small probe, which is a necessary foundation for any such multi-step extensions.
>
> > **W2**. Currently, the experiments seem to be all done on synthetic datasets. While this is useful for the uncertainty metrics, having some error-based analysis on some standard real-world time series datasets could strengthen the paper.
> >
>
> We would like to kindly point out that **this is not true** — all the experiments in section 5.2 were conducted on standard real-world datasets from the Darts and Monash collections, to establish whether the proposed method can also be used on real-world data. Our findings demonstrate that indeed, we achieve good performance also on real-world datasets, and for datasets in a similar range to the synthetic datasets we constructed there is also an encouraging level of generalisation — from the probe trained on the synthetic data to making predictions on the real data.

---

> ### Author Response · Authors · 2025-11-24
> **Rebuttal (Part 2/2)**
>
> ## **Questions**
>
> > **Q1**. If LLMs generate results autoregressively and the model does not work as well for the greedy approach, does this mean that it would still be hard to capture the results of the LLMs purely from the hidden states?
> >
>
> As we explain in l. 210-213, we believe the probe performs less well on predicting the greedy sample because the greedily decoded output is not strongly represented in the hidden states, unlike central distributional statistics such as the mean or median. We hypothesise that this is because the greedy prediction is a byproduct of the decoding procedure rather than a meaningful property of the underlying predictive distribution. Specifically, greedy decoding selects the most likely token at each step, but the resulting multi-token number is not necessarily the number with the highest overall probability mass (i.e., the mode), nor does it correspond to the mean or the median.
>
> This is consistent with downstream performance: using the greedy sample for next-step forecasting yields substantially higher MSE than using the mean or median (Table 3). Consequently, recovering the greedy prediction from hidden states is both less reliable and less useful than recovering central distributional statistics.
>
> Finally, we emphasise that in all of our experiments we do indeed recover distributional statistics of the LLM **purely from its hidden states**, and we are happy to provide further clarification if the reviewer has any remaining concerns.
>
> > **Q2**. How much contribution does the floating-point formulation actually bring for the probe, compared to just predicting the value directly? If it is significant, then how would this method compare to using it against a standard regressor (without LLM) with the same floating-point formulation?
> >
>
> We assume that in this question the reviewer is asking to what extent our magnitude-factorised probe (as described in Section 2) compares against a standard MLP probe, where the prediction of the magnitude and the scaled value is not separated. We already address this question in Table 9 of Appendix C.1, where we compare the performance of these two variants on the synthetic dataset with values in range $[-1.0, 1.0]$. The results demonstrate that using the magnitude-factorised model allows to lower the error by at least 33% even on this dataset with the narrowest range out of those we consider.
>
> Regarding the comparison with a “standard regressor,” we would appreciate clarification on what the reviewer envisions as the input to such a regressor. Our goal is specifically to recover the properties of the LLM’s predictive distribution, and therefore our probe operates on the LLM’s hidden states to elicit this information. It is unclear to us what the goal of a regressor “without LLM” would be. We would be happy to run an additional comparison once we better understand the intended comparison.
>
> > **Q3**. Why is the LLM sample error so high in the beginning for Figure 4?
> >
>
> The plot in Figure 4 shows the MSE between the the ground-truth $x_{n+1}$ and the LLM’s prediction taken as the empirical mean of $p_{\text{LLM}}(x_{n+1} \vert x_{1:n})$ estimated using $N$ samples $\\{y^j\\}\_{j=1}^N \sim p_{\text{LLM}}(x_{n+1} \vert x_{1:n})$. The initial LLM-sampling error in Figure 4 is high because, for small $N$, the empirical estimator of the mean has high variance. When $N=1$, the estimate of the mean reduces to a single stochastic sample from $p_{\text{LLM}}(x_{n+1} \mid x_{1:n})$. A single sample can easily deviate substantially from the true mean due to the inherent randomness of the LLM’s predictive distribution.  As $N$ increases, the estimator variance decreases, and the error correspondingly falls.
>
> ---
>
> We hope these clarifications addressed your concerns and we are happy to answer any outstanding questions.

---

> > ### Comment · Reviewer_hDqn · 2025-11-25
> >
> > Thanks for the author's responses, I have a clearer understanding of the paper now.
> > I apologize for the point that I made that was wrong, and have removed that from the review since it was misinformed, and adjusted the score upwards.

---

> > > ### Author Response · Authors · 2025-11-25
> > >
> > > Thank you very much for a constructive discussion, we appreciate your support of our work!

---

### Official Review · Reviewer_x9bF · 2025-11-01

**Soundness:** 3
**Presentation:** 3
**Contribution:** 3
**Rating:** 6
**Confidence:** 3

**Summary:**

The paper introduces a set of regression probes to predict the distributional properties of LLM-based time-series forecasters. By training the probes on synthetic data, the probes can predict quantities such as the greedy prediction, mean, and median with good accuracy without autoregressive decoding in both ID time-series and, to a less extent, OOD time-series.

**Strengths:**

- I appreciate the careful design going into designing the parameterization and loss functions of the probes, which is important for handling values with large ranges.
- The empirical result shows good agreement between the probe predictions and actual values obtained from decoding.
- The computational efficiency of the probe over decoding / sampling is appealing.
- Overall good presentation.

**Weaknesses:**

- A main finding of the paper is that LLM’s predictive distribution is encoded in its internal activations. This statement seems trivially true. The predictive distribution (jointly over all future tokens) is fully determined given the hidden states of the LLM (KV cache) as a direct consequence of the model architecture.
- The probes only make predictions about a single next value, rather than a future sequence. In practice, time-series forecasters are used to make extended, variable-length predictions over a non-trivial horizon, where this approach does not trivially generalize over to. In other words, the probe only shortcuts autoregression over the digits, but not time steps.
- I'm not convinced that the probes provide significant computational speedup over directly decoding from the LLM. In both approaches, most of the computation is in running the LLM forward pass on the input sequence. Decoding from the LLM is much cheaper, especially for predicting only the very next value. Thus, the savings from the probes can be negligible relative to the overall cost.

**Questions:**

- Can the proposed approach be generalized to predict distributional properties of a future sequence, rather than a single value?
- How much total runtime or compute saving do the probes provide over decoding from the LLM across the experiments?

---

> ### Author Response · Authors · 2025-11-24
> **Rebuttal (Part 1/2)**
>
> We thank the reviewer for their thoughtful and constructive feedback. We appreciate the recognition of our contribution and the soundness of our experimental design, and are grateful for the reviewer’s suggestions which helped us identify areas for clarification. Below we address all your comments point by point.
>
> ## **Weaknesses**
>
> > **W1.** A main finding of the paper is that LLM’s predictive distribution is encoded in its internal activations. This statement seems trivially true. The predictive distribution (jointly over all future tokens) is fully determined given the hidden states of the LLM (KV cache) as a direct consequence of the model architecture.
> >
>
> Thank you for raising this point. We agree that the LLM’s predictive distribution is a deterministic function of the input and its full internal state, including all network parameters (MLP weights, attention matrices, etc.).
>
> However, our key contribution is the empirical finding that the **hidden representation of the last input token alone** contains sufficient information for a **low-capacity probe** (two 1-layer MLPs with hidden size 512) to accurately recover key properties of the model’s predictive distribution over **multiple future tokens**. The probe is orders-of-magnitude smaller than the LLM and does not perform autoregressive computations or have access to the LM head, KV cache, or other model parameters.
>
> This observation is not implied by the model architecture. While the full predictive distribution is deterministically computable from the entire set of LLM's weights, this does not guarantee that
>
> 1. All relevant information is encoded in the hidden representations of the last token only,
> 2. The mapping between hidden states and statistics of the distribution is sufficiently low-dimensional or smooth to be captured by a small probe, or
> 3. The probe can recover multi-token distributional statistics without performing autoregressive computations.
>
> Our results show that the model internally organises predictive-distribution information in a structured way that enables such recoverability—an empirical property that is not guaranteed by the determinism of the predictive distribution as a function of the LLM's complete set of weights.
>
> ---
>
> > **W2.** The probes only make predictions about a single next value, rather than a future sequence. In practice, time-series forecasters are used to make extended, variable-length predictions over a non-trivial horizon, where this approach does not trivially generalize over to. In other words, the probe only shortcuts autoregression over the digits, but not time steps.
> >
>
> It is correct that our probe predicts only the next-step distribution $p(x\_{n+1} \mid x\_{1:n})$. This is fully consistent with standard practice in time-series forecasting and autoregressive modelling: many widely used models (e.g., ARIMA, state-space models) are trained to estimate the **one-step-ahead** predictive distribution, with multi-step forecasting obtained via recursive application (predict $\hat{x}\_{n+1}$, append it to the input, predict $\hat{x}\_{n+2}$, etc.). We do not claim that our method avoids autoregression over time. Rather, our contribution is that it avoids **autoregression at the token level of the next number**.
>
> Because numerical value $x_i$ is typically represented using multiple tokens, learning the one-step-ahead distribution $p(x_{n+1} \mid x_{1:n})$ still requires recovering a nontrivial distribution over multiple future tokens without performing token-level rollouts. Our probe does exactly this.
>
> We agree that extending our framework to predict distributions over multiple future values is an interesting direction for future work.  However, this pursuit faces non-trivial design choices which are dependent on the downstream application: should the probe target just the marginal distributions $p(x_{n+k} | x_1, \dots, x_{n})$ for $k \geq 1$, or also capture dependencies on the intermediate predictions $x_{n+1}, \dots, x_{n+k-1}$?  The present contribution demonstrates that next-value predictive distributions are recoverable from internal representations using a small probe, which is a necessary foundation for any such multi-step extensions.

---

> ### Author Response · Authors · 2025-11-24
> **Rebuttal (Part 2/2)**
>
> ---
>
> > **W3.** I'm not convinced that the probes provide significant computational speedup over directly decoding from the LLM. In both approaches, most of the computation is in running the LLM forward pass on the input sequence. Decoding from the LLM is much cheaper, especially for predicting only the very next value. Thus, the savings from the probes can be negligible relative to the overall cost.
> >
>
> Please see our analysis in section C.3 for a detailed breakdown and comparison of the computational cost (in terms of FLOPS) and the time required to generate the predictions using the LLM directly and using our probe. To summarise the results, running our probe to predict a single statistic of the LLM’s distribution (e.g. mean or median) requires no more than 34M FLOPS, compared with $n \times 70B$ FLOPS required to generate $n$ 5-digit numbers from the LLM (assuming each digit is tokenised separately, as in Llama-2). Note that generating a single number consisting of $m$ tokens requires $m$ forward passes through the LLM while our probe requires just a single forward pass. In terms of inference time, we show in our experiments that generating even *a single sample* with an LLM is roughly 45× slower than running the full inference pipeline with our probe (including the generation of the LLM embeddings) for a single statistic of the LLM’s distribution. When it comes to generating 20 samples (which offers comparable performance on the task of next time series value prediction as our probe, cf. Fig 4), using our probe is roughly 54x faster than autoregression. We consider these to be significant performance speed-ups.
>
> ## **Questions**
>
> > **Q1.** Can the proposed approach be generalized to predict distributional properties of a future sequence, rather than a single value?
> >
>
> See the answer to W2.
>
> > **Q2**. How much total runtime or compute saving do the probes provide over decoding from the LLM across the experiments?
> >
>
> See the answer to W3.
>
> ---
>
> We hope these clarifications addressed your concerns and we are happy to answer any outstanding questions.

---

### Official Review · Reviewer_Mkh9 · 2025-11-06

**Soundness:** 4
**Presentation:** 4
**Contribution:** 3
**Rating:** 8
**Confidence:** 3

**Summary:**

This paper addresses the question of whether LLM's hidden representations encode distributional information about their numerical predictions rather than relying on autoregressive decoding when it comes to utilizing LLMs for time-series forecasting regression tasks. They Introduce a probing model that separately predict the order of magnitude ( treated as classification ) and the multiplicative residual to predict the mean, median and the greedy predictions from the LLM internal representations. They further extend their results to predicting multiple quantiles of the underlying distribution utilizing quantile regressors. They lastly validate generalization capabilities of the underlying predictors.

**Strengths:**

1. the paper is very well written. All experiments all well-motivated, clearly explained, and results are well presented.

2. The idea of eliciting numerical predictions and uncertainties from hidden states is timely, and can possible be extended to tasks beyond time series forecasting.

3. The experiments are sound, and different aspects such as different kind of generalizations are well considered, thus are comprehensive. I found the calibration analysis interesting extension of the experiments and that it strengthens the credibility of the results.

Lastly, and to summarize, the paper is well written, and the findings are interesting, informative, and of interest to a broader community. The evidence that distributional information about LLM predictions are recoverable from hidden states is enlightening followed by sound experimental evidence.

**Weaknesses:**

Most of the weaknesses are well acknowledged in the paper. These include (i) accessing hidden states may decrease the practicality of the method, (ii) relying on extensive autoregressive sampling for training the probes.

1. While the paper positions itself only within the context of regression and time-series forecasting, it does not sufficiently discuss prior work that also show hidden states encode future outputs and uncertainty in other modalities and tasks. There is a growing literature that shows similar behaviors that probe internal states in the context of analyzing sentiment, factuality, intent, jailbreaking, chain-of-thought reasoning and etc. The related works would benefit from addressing these works as well for better positioning the contribution within a larger context.

2. The experiments in section 2.2 focus on [-1,1] range but its not very clear how the method behaves for larger value ranges. although the authors mention testing other ranges, the results don't show how performance and calibration changes with scale, and it would be interesting to see how performance metric across different ranges change; whether larger numeric ranges introduce instability, or if normalization essentially removes that issue.

**Questions:**

1. How sensitive are the probe's predictions to the choice of layers used ? It would be interesting to analyze which layers contribute the most to recovering the numerical predictions, and whether the same choice of "best" layers is consistent across tasks and different settings studied in the paper under "generalization".

2. One form of generalization that is not studied in the experiments is whether a probe trained on llama2 for example generalizes ( or not ) to different models from the same family ( for example llama3 ). Results discussing this would be interesting.

---

> ### Author Response · Authors · 2025-11-24
>
> We thank the reviewer for their thoughtful and constructive feedback. We appreciate the recognition of our contribution and the soundness of our experimental design, and are grateful for the reviewer’s suggestions which helped us identify areas for clarification and prompted to conduct additional experiment. Below we address all your comments point by point.
>
> ## Weaknesses
>
> > **W1.** While the paper positions itself only within the context of regression and time-series forecasting, it does not sufficiently discuss prior work that also show hidden states encode future outputs and uncertainty in other modalities and tasks. There is a growing literature that shows similar behaviors that probe internal states in the context of analyzing sentiment, factuality, intent, jailbreaking, chain-of-thought reasoning and etc. The related works would benefit from addressing these works as well for better positioning the contribution within a larger context.
> >
>
> Thank you for this suggestion. Following your comment, we have now included an additional section in our Related Works discussing what information can be recovered directly from the hidden states of the LLMs across different data modalities, and explaining the unique contributions of our work in the context of these findings.
>
> > **W2.** The experiments in section 2.2 focus on [-1,1] range but its not very clear how the method behaves for larger value ranges. although the authors mention testing other ranges, the results don't show how performance and calibration changes with scale, and it would be interesting to see how performance metric across different ranges change; whether larger numeric ranges introduce instability, or if normalization essentially removes that issue.
> >
>
> The results we present in Section 2.2 consist of two parts: in Table 1, we indeed focus only on datasets in range $[-1, 1]$ to make the comparison using MSE more meaningful. However, in Figure 2, we demonstrate results obtained on the large concatenated dataset with values in range  $[-10000, 10000]$. Specifically, we plot the predicted vs. true values of the mean, median and greedy predictions, alongside a bar chart showing the accuracy in predicting the magnitude of the point predictions. The results indicate that our method remains robust and gives accurate results also for datasets of larger scale. We acknowledge that it was not clear that Figure 2 is based on the dataset with the larger range, and we have now added a note in Section 2.2  clarifying this. Thank you for raising this point.
>
> ## Questions
>
> > **Q1.** How sensitive are the probe's predictions to the choice of layers used ? It would be interesting to analyze which layers contribute the most to recovering the numerical predictions, and whether the same choice of "best" layers is consistent across tasks and different settings studied in the paper under "generalization".
> >
>
> Thank you for this question, we also find it intriguing to see which layers contain the information necessary to recover the future predictions. Appendix C.4 of our submission already includes the results of an ablation study aiming to identify which layers contain most information about both the point predictions (sample mean, greedy prediction) and the uncertainty (quantiles). The results demonstrate that the point predictions (particularly the mean) are mostly encoded in the last layers of the LLMs, whereas uncertainty is encoded more uniformly across the layers.
>
> > **Q2.** One form of generalization that is not studied in the experiments is whether a probe trained on llama2 for example generalizes ( or not ) to different models from the same family ( for example llama3 ). Results discussing this would be interesting.
> >
>
> This is an interesting point! We have explored this question in our preliminary experiments, but found this form of generalisation to not hold. To demonstrate some quantitative results, we have evaluated our probe trained on the hidden representations of Llama-3 and tested on hidden representations generated by Llama-2 (dataset with values in range $[-1, 1]$). The resulting MSE was: $66.75$ for greedy predictions, $163.29$ for mean and $45.71$ for median. Comparing these values with those presented in Table 1 (where the errors fall below $0.01$), we see that there is no sign of generalisation from one model to another, even within the same family of LLMs.

---

### Meta-Review · Area_Chair_tCpZ · 2026-01-13

**Summary:**

The paper proposes regression probes that directly predict statistics of an LLM regressor's predictive distribution from its internal hidden states, without explicit autoregressive decoding. The reviewers found the paper well written, timely, and the experiments sound and strong. The reviewers raised concerns about the main claim (i.e., that a LLM's prediction is encoded in its internal state) "trivially true", that only one-step prediction is explored, missing related work, a lack of evidence for speedup claims, a lack of theoretical or architectural explanations, and that the uncertainty part could be strengthened.

**Reviewer Concerns:**

In the rebuttal, the authors address the missing related work, provide analysis for speedup claims, and clarify architectural aspects with reference to an ablation study.

**Reviewer Scores:**

Reviewer hDqn would likely increase the score after the clarifications in the rebuttal (and the reviewer also directly states this). There is a slight chance that review 8jM1 would increase the scores, based on the ablation study. Reviewers Mkh9 and x9bF would likely remain unchanged in their positive stance. The initial scores already indicate acceptance, and assuming Review hDqn increases the score, the paper is clearly accepted. Therefore, I recommend accepting the paper.

---

### Decision · Program_Chairs · 2026-01-26

Accept (Poster)